Subject Areas:
nanotechnology

Keywords:
gold nanoparticles, sedimentation–diffusion equilibrium, settling dynamics, gravitational sedimentation, target dose

Author for correspondence:
Francesco Giorgi
e-mail: francesco.giorgi@liverpool.ac.uk

# Settling dynamics of nanoparticles in simple and biological media

Francesco Giorgi[1], Peter Macko[2], Judith M. Curran[1], Maurice Whelan[2], Andrew Worth[2] and Eann A. Patterson[1]

[1]School of Engineering, University of Liverpool, Brownlow Hill, Liverpool L69 3BX, UK
[2]European Commission, Joint Research Centre, Via E. Fermi 2749, 21027 Ispra, Italy

 FG, 0000-0002-7550-4217; EAP, 0000-0003-4397-2160

The biological response of organisms exposed to nanoparticles is often studied *in vitro* using adherent monolayers of cultured cells. In order to derive accurate concentration–response relationships, it is important to determine the local concentration of nanoparticles to which the cells are actually exposed rather than the nominal concentration of nanoparticles in the cell culture medium. In this study, the sedimentation–diffusion process of different sized and charged gold nanoparticles has been investigated *in vitro* by evaluating their settling dynamics and by developing a theoretical model to predict the concentration depth profile of nanoparticles in solution over time. Experiments were carried out in water and in cell culture media at a range of controlled temperatures. The optical phenomenon of caustics was exploited to track nanoparticles in real time in a conventional optical microscope without any requirement for fluorescent labelling that potentially affects the dynamics of the nanoparticles. The results obtained demonstrate that size, temperature and the stability of the nanoparticles play a pivotal role in regulating the settling dynamics of nanoparticles. For gold nanoparticles larger than 60 nm in diameter, the initial nominal concentration did not accurately represent the concentration of nanoparticles local to the cells. Finally, the theoretical model proposed accurately described the settling dynamics of the nanoparticles and thus represents a promising tool to support the design of *in vitro* experiments and the study of concentration–response relationships.

## 1. Introduction

A dose–response assessment describes the response of an organism as a function of exposure to an exogenous agent. In the

pharmaceutical sector, for example, dose–response analysis is essential to estimate the dose at which a drug can have a therapeutic or adverse effect in patients [1]. One of the main principles of the dose–response paradigm is that the elicited biological response is proportional to the concentration of an administered agent at the site of action [2]. Thus, knowing the concentration within the target organ, tissue or cell population is a prerequisite for accurate dose–response modelling. However, measuring the local concentration at the site of action is not always practicably possible and thus often the initial administered concentration is used, which can lead to inaccuracies in determining dose–response relationships [3].

Gold nanoparticles are of great interest for a number of biomedical applications since they are a promising carrier for the targeted delivery of therapeutic, diagnostic and imaging agents in the human body [4]. They are also being explored as a means of subjecting stem cells to mechanical stimuli to activate certain signalling pathways and to modulate their differentiation [5]. As with small-molecule agents, nanoparticle concentration has been reported as one of the primary factors determining positive as well as adverse effects on the biological organisms tested [6,7].

*In vitro* studies of nanoparticles typically involve exposing a monolayer of cultured cells that are adhered to the bottom of a well in a microtiter plate. Frequently investigators use the nominal concentration of nanoparticles in the cell medium as a basis to determine dose–response relationships without taking into account that the concentration may not in fact be homogeneous in the well over time. This is due to the fact that nanoparticles in solution are subject to gravitational sedimentation forces, Brownian diffusion forces and inter-particle forces which affect the transport of nanoparticles within the medium. This in turn controls the concentration of nanoparticles local to the cells that can influence cellular uptake and toxicity [8,9]. Krug, in an effort to highlight shortcomings in the knowledge of nanosafety, suggested that nanotoxicological studies should include, as a requirement, a consideration of the appropriate dose and/or concentration at the cellular level and the inclusion of a dose–effect relationship in the study design [10]. The aim of this study is to provide a better understanding of the local concentration of nanoparticles to which a monolayer of cells is exposed by characterizing in real time the transport of nanomaterial through the solution over time.

## 1.1. Sedimentation–diffusion equilibrium

Nanoparticles dispersed in solution tend to sediment, diffuse and aggregate as a function of intrinsic properties (size, mass, surface charge, etc.) and as a function of system properties (viscosity, temperature, etc.). Once injected in solution, the process dominating the dynamics of nanoparticles is the diffusion process and nanoparticles tend to move from zones of high concentration to zones of low concentration (figure 1a). After a certain time, an unstable equilibrium is achieved, and nanoparticles are randomly but homogeneously distributed in the medium (figure 1b). Gravitational sedimentation becomes relevant for dense particles (such as metallic particles) with a diameter larger than 10 nm, causing a concentration gradient from the bottom to the top of the solution [11]. This concentration gradient causes the diffusion flux of the particles to be in the opposite direction to their sedimentation flux. When the diffusion flux equals the sedimentation flux, a so-called sedimentation–diffusion equilibrium is achieved (figure 1c) [12]. The time at which this dynamic equilibrium is reached is called the settling time.

Most studies described in the literature attempt to experimentally investigate the sedimentation–diffusion equilibrium process by acquiring UV–visible absorption spectra over time in an attempt to determine the behaviour of the nanoparticles in solution [13–15]. UV–visible spectroscopy is a powerful technique mainly used to analyse the colloidal stability of nanoparticle suspensions. The absorbance peak of the UV spectrum is related to the concentration of monodispersed particles in solution [16]. This technique has been used to investigate the sedimentation process by evaluating the decrease in the concentration of monodispersed nanoparticles in solution over time. However, the detection limit of the spectrophotometer makes it impossible to perform analysis at low concentrations of particles [17]. Moreover, aggregates of particles can float in solution for a certain time depending on their size and the depth at which they were formed [18]. Finally, uncertainty in the acquired UV–visible spectra has been reported for solutions exhibiting a high gradient of nanoparticle concentration from the bottom to the top of the acquisition zone [14].

In this study, we investigated the sedimentation–diffusion process and evaluated the settling time of gold nanoparticles in a range of simple and biological solutions. Nanoparticles were tracked by exploiting the optical phenomenon of caustics, and the nanoparticle concentration profile was directly evaluated by acquiring consecutive optical sections from the bottom to the top of the solution and counting the number of particles in each section over time.

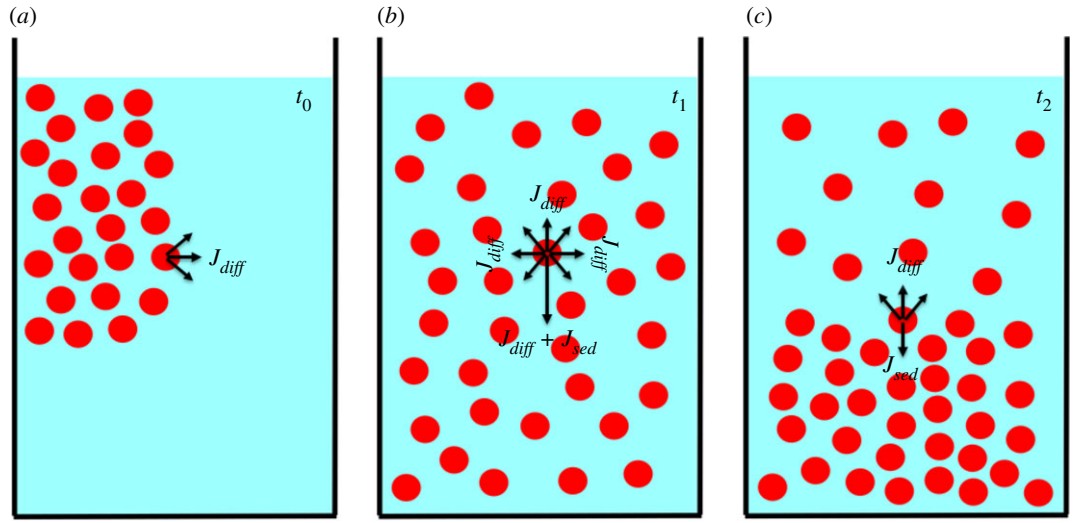

**Figure 1.** Schematics of the three phases of sedimentation–diffusion equilibrium for nanoparticles in solution. (*a*) Once injected in solution nanoparticle dynamics is mainly regulated by diffusion forces. Nanoparticles are dispersed over the entire available volume. (*b*) Nanoparticles are randomly but homogeneously dispersed in solution. Gravity tends to direct particles to the bottom of the solution. (*c*) Sedimentation–diffusion equilibrium is attained. The rate of transport of nanoparticles in any one direction due to sedimentation equals the rate of transport in the opposite direction due to diffusion.

Investigators have also developed theoretical models to predict the observed sedimentation–diffusion behaviour of nanoparticles in solution [19]. The most common theoretical model used to describe the concentration profile of nanoparticles in solution is the well-known Mason–Weaver equation, which combines the sedimentation and diffusion processes to predict the concentration of a solute at a certain depth in a solution [20]. The parameters of the Mason–Weaver equation are the diffusion coefficient and the sedimentation velocity calculated using the Stokes–Einstein diffusion equation and the Stokes sedimentation law, respectively. The Mason–Weaver model is based on the assumption that there are no interactions between nanoparticles and that their motion is only governed by random Brownian forces and a directional gravitational force [21]. However, the assumption of non-interacting particles can be valid only for a system with a very low particle concentration. Moreover, the Stokes–Einstein diffusion equation and Stokes sedimentation law can fail when predicting the dynamic behaviour of particles at the nanoscale because they do not take into account factors which have been reported to be dominant at the nanoscale, such as electrostatic and van der Waals forces [22]. Several studies have already demonstrated that, depending on the properties of the nanoparticles and solution, the experimental diffusion and sedimentation coefficients differ from theoretical ones at the nanoscale [23–25].

In this study, we investigated the settling dynamics of different sized gold nanoparticles in water and cell culture medium at a range of different temperatures by exploiting the optical phenomenon of caustics [26,27]. The comparison between experimental data and theoretical predictions of the Mason–Weaver equation confirmed that the current theoretical framework based on the Stokes–Einstein diffusion and Stokes sedimentation equations is not able to accurately describe the real dynamics of nanoparticles settling in a solution. Here, we propose a modified version of the Mason–Weaver equation able to better predict the experimental concentration depth profile of nanoparticles, by making the diffusion coefficient a function of the local concentration of nanoparticles, which is a factor that has been reported to influence the dynamics of nanoparticles in solution [24].

# 2. Experimental

## 2.1. Material and methods

Spherical, citrate-capped, negatively charged gold nanoparticles (pH 7.7) were purchased from BBI Solutions, with nominal diameters of 60 nm, 80 nm, 100 nm and 150 nm. Spherical positively charged gold nanoparticles were purchased from nanoComposix, with a nominal diameter of 100 nm. The as-supplied concentrations were reduced by adding the concentrate to ultrapure-deionized 0.2 µm membrane filtered water as appropriate to obtain a constant working concentration of

$10^8$ particles ml$^{-1}$. Nanoparticle dynamics in solution were monitored in a standard optical inverted microscope (Axio Observer.Z1 m, Carl Zeiss) mounted on antivibration feet (VIBe, Newport) to isolate the sample from the environment. The microscope was equipped with a stage-top incubation system (Incubator PM S1, Heating Insert P S1, Temp and CO2 module S1, Carl Zeiss) to control the temperature and the amount of $CO_2$ present during the experiments. To generate caustic signatures of the nanoparticles, some simple adjustments to the normal set-up of the microscope were made following the procedure described by Patterson & Whelan [26]. The caustic curves generated are several orders of magnitude larger than the real nanoparticles in solution, allowing their detection in an optical microscope without any requirement for fluorescent labelling (electronic supplementary material, figure S2a). The optical set-up proposed by Patterson and Whelan has been proven to be able to generate caustics signatures of nanoparticles of a variety of materials (gold, silica, polystyrene) and with a diameter as small as 3 nm [26,27]. The shape of the caustics signatures also allows easy differentiation between monodispersed spherical particles and aggregates/clusters in solution at low concentrations of particles [27]. The concentration profiles of the nanoparticles in solution were evaluated using 60 µl of nanoparticle solution in a 250 µm ± 10 µm deep cavity in a microscopy slide and acquiring z-stacks from the bottom to the top of the solution. The distance between consecutive images was set equal to 4 µm for a total acquisition length of 250 µm ± 10 µm. The value of 4 µm between consecutive acquisitions was chosen to avoid multiple counting of the same particles (caustics signatures extend for less than 1 µm in length along with the z-direction [27]). The number of particles in each image was counted using the software ImageJ and integrated from the mid-depth (125 µm ± 10 µm from the bottom) to the top (250 µm ± 10 µm from the bottom) of the solution to obtain the total number of nanoparticles not settled at each time step (electronic supplementary material, figure S2b). The depth of 125 µm was chosen because experimental observations demonstrated that particles formed a sediment that extended over the bottom half of the solution. Nanoparticles in the bottom half and in the top half of the solution have been identified, for the sake of simplicity, as settled and not settled, respectively, because the term 'settle' is commonly used in the scientific community to identify particles forming a sediment. The full concentration profile is not provided because, as the concentration of the nanoparticles forming a sediment increases, the caustics in the bottom half of the solution overlap, making measurements in that section less precise and unreliable. Four solutions were prepared separately for each sedimentation test so that the results presented are average values with standard deviations. Each test was started 5 min after the injection of nanoparticles into the solution, to guarantee consistency between experiments and to allow nanoparticles to distribute throughout the medium uniformly but randomly. The concentration of nanoparticles not settled has been normalized so that the data presented are in the range 0–1. The settling time of each nanoparticle solution was established by evaluating the time at which the gradient of concentration of nanoparticles not settled over time was less than 5%. The UV–visible spectra shown in figures 5 and 6 were acquired with a UV–visible spectrophotometer (U-2900, Hitachi) using 3 ml of the solution of nanoparticles in a 45 mm deep glass cuvette.

## 2.2. Mason–Weaver equation

The objective of the predictive analysis was to calculate the height profile of the nanoparticle concentration in the solution as a function of time. It was achieved by solving the Mason–Weaver convection–diffusion differential equation [20]:

$$\frac{\partial n(z,t)}{\partial t} = D\frac{\partial^2 n(z,t)}{\partial z^2} - V\frac{\partial n(z,t)}{\partial z}, \tag{2.1}$$

where $n(z,t)$ is the normalized nanoparticle concentration, $D$ is the diffusion coefficient and $V$ is the sedimentation velocity (in our coordinate system the direction of the sedimentation velocity is the opposite to the z-axis orientation so $V < 0$ in all equations). The initial condition was $n = 1$ for all $z$ and $t = 0$. The boundary condition was based on an assumption of nanoparticle conservation in the system, expressed by no particle flux across the top and the bottom of the solution (at $z = 0$ and $L$ being the depth of the solution).

## 2.3. Mason–Weaver equation with variable diffusion coefficient

Following the assumption that the diffusion coefficient $D$ is not constant but depends on the local concentration of the nanoparticles and thus also on the position $z$ and time $t$, the Mason–Weaver

convection–diffusion differential equation in the form of (2.1) is not correct and needs to be derived supposing a function of the form $D(n(z,t))$. We can start with Fick's second law:

$$\frac{\partial n(z,t)}{\partial t} = -\frac{\partial J(z,t)}{\partial z},$$

(2.2)

where $J(z,t)$ is the flux of nanoparticles. In our scenario:

$$J(z,t) = J_D(z,t) + J_S(z,t),$$

(2.3)

where $J_D(z,t)$ is the diffusion flux defined by Fick's first law:

$$J_D(z,t) = -D(n(z,t))\frac{\partial n(z,t)}{\partial z},$$

(2.4)

and $J_S(z,t)$ is the sedimentation flux expressed as

$$J_S(z,t) = Vn(z,t).$$

(2.5)

Combining (2.2) to (2.5), we get the Mason–Weaver equation with a non-constant diffusion coefficient, $D$:

$$\frac{\partial n(z,t)}{\partial t} = \frac{\partial}{\partial z}\left(D(n(z,t))\frac{\partial n(z,t)}{\partial z}\right) - V\frac{\partial n(z,t)}{\partial z}.$$

(2.6)

To solve it, the $z$ coordinate was first transformed to a non-dimensional form:

$$z' = \frac{z}{L}.$$

(2.7)

Then equation (2.6) was solved using the Euler explicit method where the derivatives were approximated by finite differences. The space and time domains were uniformly partitioned in a mesh. The space step was $L/100$ so the non-dimensional $\Delta z = 0.01$, and the time step $\Delta t = 0.4\,\text{s}$ was chosen to maintain a stable solution of the equation over time.

The points $n(z'_j, t_u) = n_j^u$ represent the numerical approximation of the concentration at the space point $z'_j$ and time point $t_u$. The equation (2.6) after the approximation by finite differences becomes

$$\frac{n_j^{u+1} - n_j^u}{\Delta t} = \frac{D(n_{j+1}^u)(n_{j+1}^u - n_j^u) - D(n_j^u)(n_j^u - n_{j-1}^u)}{L^2 \Delta z^2} - \frac{V(n_{j+1}^u - n_j^u)}{L\Delta z}.$$

(2.8)

Using this relation and knowing the values of the concentration at time $u$, we can obtain the concentration at time $u+1$ (computational stencil shown in figure 2a):

$$n_j^{u+1} = n_j^u + \Delta t \left( \frac{D(n_{j+1}^u)(n_{j+1}^u - n_j^u) - D(n_j^u)(n_j^u - n_{j-1}^u)}{L^2 \Delta z^2} - \frac{V(n_{j+1}^u - n_j^u)}{L\Delta z} \right).$$

(2.9)

To calculate the nanoparticle concentration at the boundaries, we apply the boundary condition that there is no particle flux across the top and the bottom of the solution. By combining equations (2.2) and (2.3), the Mason–Weaver equation is obtained as a function of the nanoparticle fluxes:

$$\frac{\partial n(z,t)}{\partial t} = -\frac{\partial}{\partial z}(J_D(z,t) + J_S(z,t)).$$

(2.10)

The derivatives in this equation (2.10) can be approximated by finite differences:

$$\frac{n_j^{u+1} - n_j^u}{\Delta t} = -\frac{1}{L}\frac{J_D'' - J_D'}{\Delta z} - \frac{1}{L}\frac{J_S'' - J_S'}{\Delta z},$$

(2.11)

where $J_D''$ and $J_S''$ are the diffusion and sedimentation fluxes between $n_j^u$ and $n_{j+1}^u$, and $J_D'$ and $J_S'$ are the diffusion and sedimentation fluxes between $n_{j-1}^u$ and $n_j^u$.

At the top boundary, $z = L$ ($j = 100$), these four fluxes, depicted also in the computational stencil (figure 2b), are as follows:

$$J_D'' = 0, \ J_D' = -\frac{D(n_{100}^u)}{L}\frac{(n_{100}^u - n_{99}^u)}{\Delta z}$$

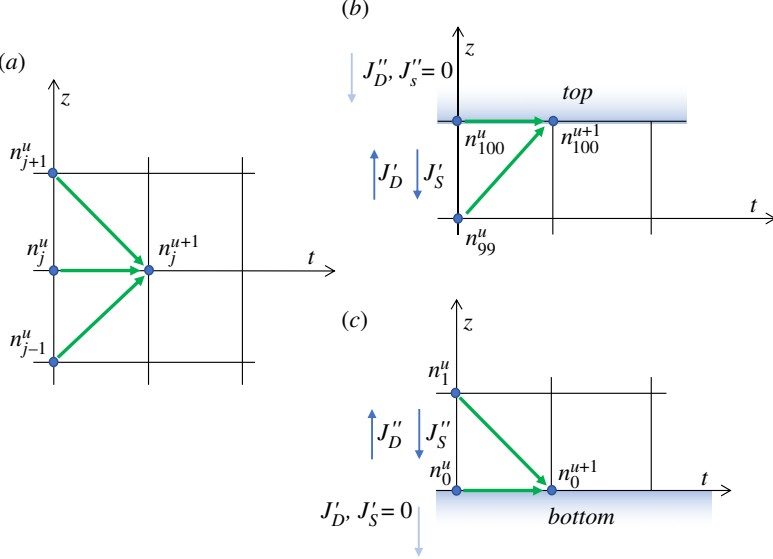

**Figure 2.** The computational stencils of the explicit Euler forward method (*a*), and at the top (*b*) and at the bottom (*c*) with depicted fluxes.

and

$$J_S'' = 0, \ J_S' = V n_{100}^u.$$

Consequently, the equation (2.11) at the top boundary takes the form

$$n_{100}^{u+1} = n_{100}^u + \Delta t \left( -\frac{D(n_{100}^u)}{L^2} \frac{(n_{100}^u - n_{99}^u)}{\Delta z^2} + \frac{V}{L} \frac{n_{100}^u}{\Delta z} \right). \tag{2.12}$$

Similarly, at the bottom boundary, $z = 0$ ($j = 0$), the four fluxes $J_D''$, $J_S''$, $J_D'$ and $J_S'$ (figure 2*c*) are as follows:

$$J_D'' = -\frac{D(n_1^u)}{L} \frac{(n_1^u - n_0^u)}{\Delta z}, \ J_D' = 0$$

and

$$J_S'' = V n_1^u, \ J_S' = 0.$$

Consequently, the equation (2.11) at the bottom boundary becomes

$$n_0^{u+1} = n_0^u + \Delta t \left( \frac{D(n_1^u)}{L^2} \frac{(n_1^u - n_0^u)}{\Delta z^2} - \frac{V}{L} \frac{n_1^u}{\Delta z} \right). \tag{2.13}$$

The diffusion coefficient $D(n)$ as a function of the normalized local concentration of the nanoparticles $n$ can be expressed as

$$D(n) = D_{\exp} + \frac{(D_{StEin} - D_{\exp}) n^k}{n_{50}^k + n^k}, \tag{2.14}$$

where $D_{\exp} = 5 \times 10^{-13} \ \mathrm{m^2 \, s^{-1}}$ is the experimental value of the diffusion coefficient measured at a low concentration of nanoparticles in the solution at 23°C and $D_{StEin}$ is the diffusion coefficient according to the Stokes–Einstein equation [24]. The transition between the diffusion coefficient at low concentrations and the value from the Stokes–Einstein relationship is regulated by the experimental fitting parameter $n_{50} = 1.5$ and $k = 30$, representing the nanoparticle relative concentration normalized at which the transition happens and the slope of the transition, respectively.

The modified version of the Mason–Weaver equation is used in this paper to theoretically predict only the settling dynamics of nanoparticles in solutions at 23°C because of the lack of data concerning the experimental diffusion coefficient exhibited by nanoparticles in solution at higher temperatures.

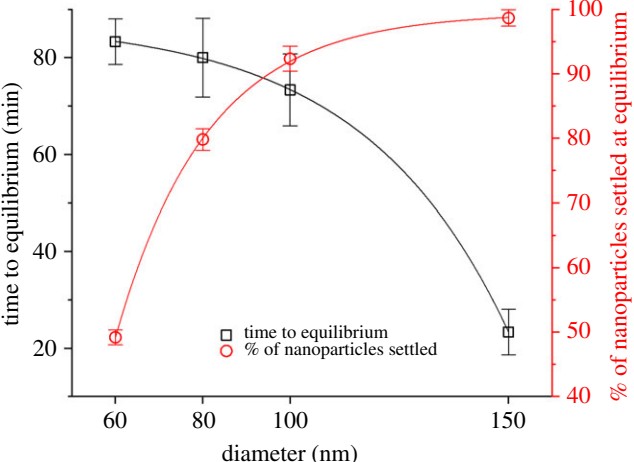

**Figure 3.** Experimental settling time (black squares and left axis) and percentage of nanoparticles settled at the equilibrium (red circles and right axis) as a function of nanoparticle diameter.

# 3. Results and discussion

## 3.1. Effect of size on nanoparticle sedimentation–diffusion equilibrium

The effect of nanoparticle size on the equilibrium of the sedimentation–diffusion process is shown in figure 3. The estimated concentration of settled nanoparticles and the relative settling time are both functions of the size of the nanoparticles. Larger particles tend to reach equilibrium faster with a higher concentration of particles transported to the bottom of the solution, but all the particles tested exhibit a non-negligible deviation between the administered dose and the target dose at the equilibrium. Consider, for example, the smallest size tested: approximately 50% of the 60 nm nanoparticles in solution settle and migrate towards the bottom of the solution, meaning that the target dose is at least 1.5 times the initial administered dose.

The reliability of the classical Mason–Weaver equation in describing the concentration profile of the nanoparticles in solution over time has been assessed by plotting the concentration of not-settled nanoparticles together with the Mason–Weaver predictions (figure 4a). As expected, the Mason–Weaver equation, with the Stokes–Einstein diffusion coefficient and Stokes sedimentation velocity, is not able to accurately predict the experimental results. The theoretical upward diffusion flux, which opposes the downward sedimentation flux of nanoparticles, is higher than the experimental one, leading to an overestimation of the concentration of the nanoparticles not settled once equilibrium is attained. The main factor affecting the diffusion flux in the Mason–Weaver equation is the diffusion coefficient evaluated using the Stokes–Einstein equation. Coglitore *et al.* [24] found by tracking single monodispersed nanoparticles that, below a critical size of 150 nm diameter and a critical concentration of about $10^8$ particles ml$^{-1}$, the diffusion coefficient of the nanoparticles was at least one order of magnitude smaller than the theoretical value predicted by the Stokes–Einstein equation and was independent of nanoparticle size.

Another limitation of the model is the assumption that the sedimentation velocity and diffusion coefficient exhibited by the particles in solution are constant with time, without taking into account that the concentration gradient formed once equilibrium is achieved can directly affect the sedimentation and diffusion behaviour of the nanoparticles. Ganguly & Chakraborty [28] demonstrated that the sedimentation velocity decreases as a function of the concentration because of the increased magnitude of the hydrodynamic interactions between particles. Kourki & Famili [29] observed the same decreasing trend of sedimentation velocity when investigating the sedimentation of silica nanoparticles. Coglitore *et al.* [24] explained the low values of diffusion coefficient obtained from experiments by considering that, at low concentrations, nanoparticle motion is almost entirely controlled by the diffusive regime of the fluid molecules and the particle–particle interaction can be neglected. An extensive number of studies have demonstrated that the diffusion of fluid molecules, such as water, does not satisfy the assumptions underlying the Stokes–Einstein relationship and is better described by a fractional relationship [30,31]. On the other hand, Koening *et al.* [32] reported a higher diffusion coefficient (of the same order of magnitude as the Stokes–Einstein prediction) using the same nanoparticles and solution but with a higher particle concentration. Considering this evidence, it is reasonable to conclude that the

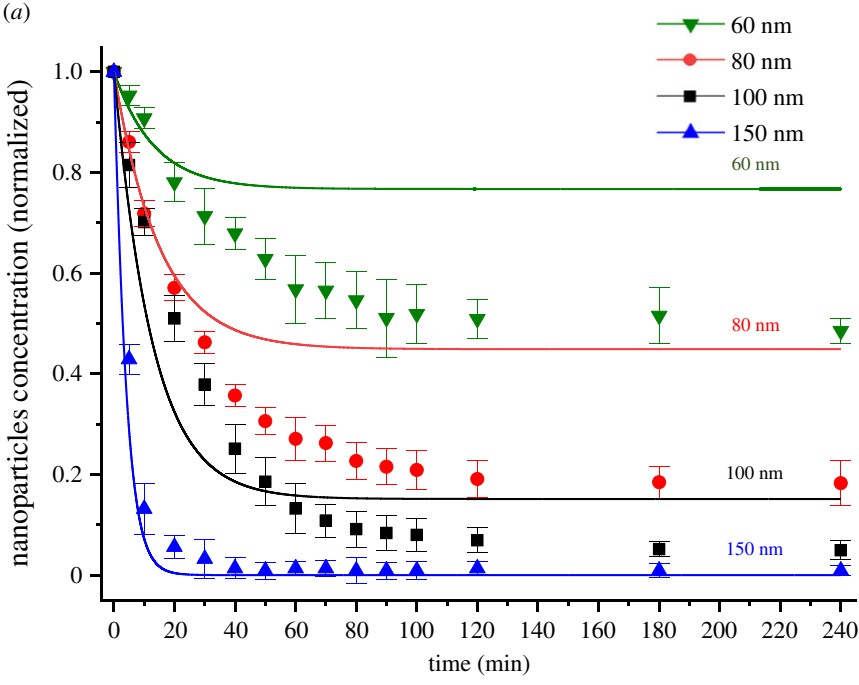

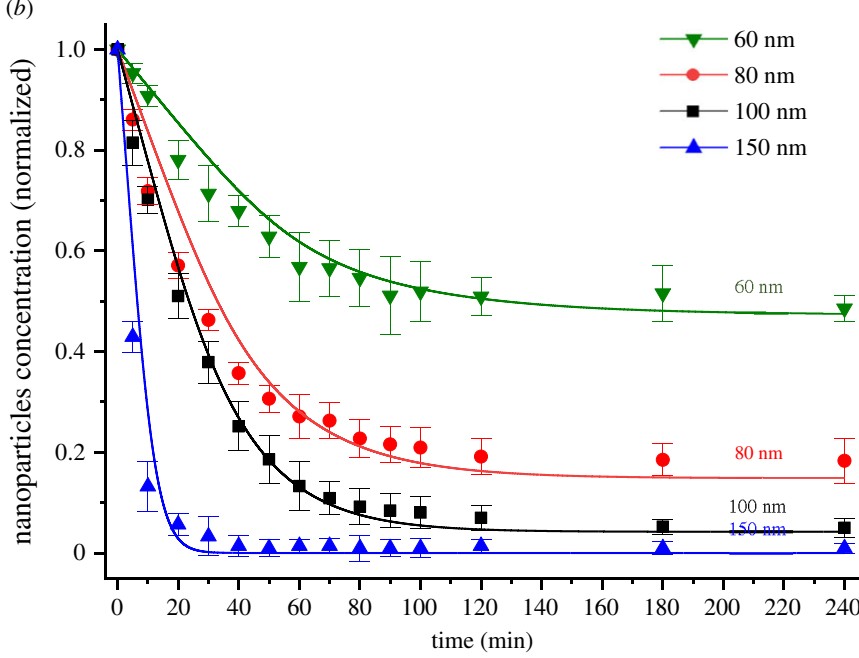

**Figure 4.** Measured concentration (symbols) of not-settled negatively charged gold nanoparticles as a function of time in water at 23°C together with predictions (lines) from (a) the Mason–Weaver model and (b) the proposed modified Mason–Weaver model.

concentration of nanoparticles directly affects their diffusion and sedimentation behaviour through particle–particle interactions. In our test scenario, at the beginning of the experiment, nanoparticles are homogeneously monodispersed at low concentrations, so that particle–particle interactions are negligible and do not affect the sedimentation velocity while the diffusion coefficient is smaller than the theoretical one and mainly regulated by the fluid molecules. Once a concentration gradient forms in the solution, particle interactions start to influence the dynamics of the nanoparticles above the settling zone, leading to a decrease in sedimentation velocity and an increase in diffusion coefficient. Another factor to take into account is the viscosity of the medium which can potentially influence both the diffusion and sedimentation behaviour of nanoparticles. However, experimental and theoretical observations have demonstrated that the presence of nanoparticles at low concentrations does not significantly affect the viscosity of the fluid [24].

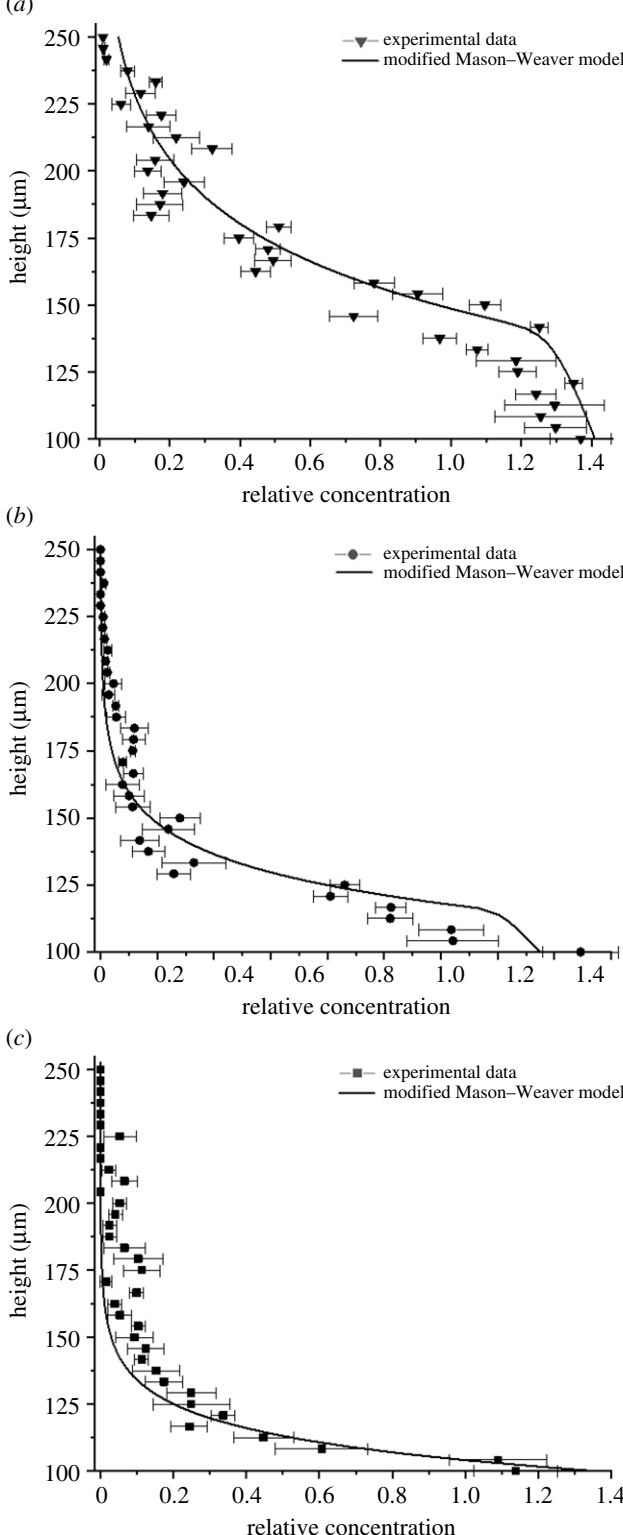

**Figure 5.** Measured concentration (symbols) of (*a*) 60 nm; (*b*) 80 nm; (*c*) 100 nm negatively charged gold nanoparticles as a function of the height of the solution in water at 23℃ after 240 min together with predictions (lines) from the proposed modified Mason–Weaver model. The concentration at each position represents the relation between the final concentration and the initial concentration (time = 0 min) measured at each specific height position. Data for 150 nm nanoparticles are not provided because both the experimental measurements and the theoretical model predictions are equal to 0 in the upper section of the solution at the equilibrium. The full concentration profile is not provided because, as the concentration of the nanoparticles forming a sediment increases, the caustics overlap, making measurements in that section less precise and unreliable.

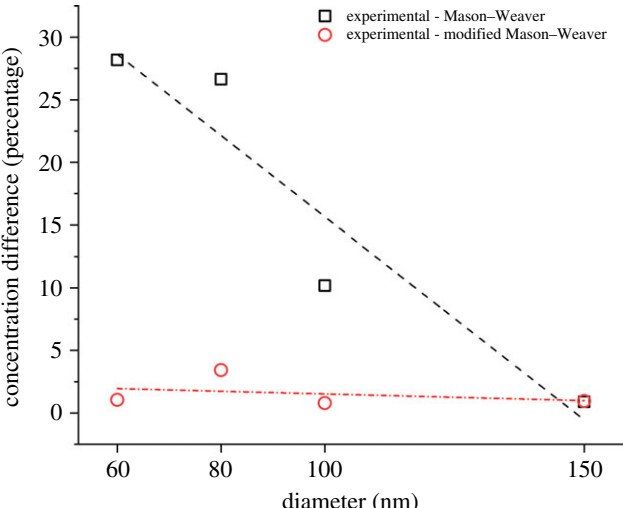

**Figure 6.** Difference between the concentration of not-settled nanoparticles at equilibrium as measured experimentally and calculated with the original and modified version of the Mason–Weaver equation.

A modified Mason–Weaver theoretical model has been developed to overcome the limitations of the classical Mason–Weaver equation, taking into account the experimental evidence and physical hypotheses discussed above. The distinctive feature of the model proposed in this study is the dependence of the diffusion coefficient on the local concentration of the nanoparticles (equation (2.6)). The diffusion coefficient for each particle size tested has been set equal to the experimental value of $5 \times 10^{-13}\,\mathrm{m^2\,s^{-1}}$, as reported in previous studies for gold nanoparticles at concentrations less than or equal to $10^8$ particles ml$^{-1}$ [22–24], and then progressively increased as a function of the local concentration of nanoparticles to the values calculated using the Stokes–Einstein equation. As shown in figure 4b, the modified model predicts the experimental data more accurately than the classical Mason–Weaver model and can be used to simulate the settling dynamics and estimate the nanoparticle concentration over time. The model presented in this work is also able to approximate the concentration profile of nanoparticles in solution. As shown in figure 5, the settling trend of the experimental data follows the theoretical predictions of the modified Mason–Weaver equation. Figure 6 provides further support for our hypothesis about the role of the diffusion of nanoparticles in driving their settling dynamics. It can be seen that the deviation between the experimental settling data and the original Mason–Weaver predictions decreases as a function of the size of nanoparticles, becoming negligible for the 150 nm gold nanoparticles tested. As the size of nanoparticles increases, the experimental diffusion coefficient has been found to be progressively better approximated by the Stokes–Einstein equation even at low concentrations [24]. Moreover, as the size of the nanoparticles increases, the sedimentation dynamics becomes the dominant factor regulating the overall dynamics, making the contribution from diffusion negligible. On the other hand, this size-dependent accuracy of the original Mason–Weaver model is not present in the modified version proposed here, where the diffusion coefficient has been made a function of the local concentration of the nanoparticles.

The minor deviations between the modified Mason–Weaver model and the experimental data can be explained by the fact that the sedimentation velocity in the model has been kept constant at a value that gives the best fit for each size of nanoparticle tested and did not vary with their concentration (electronic supplementary material, figure S1). The sedimentation velocity was kept constant to reduce the mathematical complexity and computational requirements of the modified model, but also because of the lack of experimental measurements. The latter was due to the impossibility of measuring the sedimentation process independently from the diffusion process at different time steps with the technique used in this study. Moreover, the original Mason–Weaver equation does not take into account forces that are considered to be dominant at the nanoscale (such as van der Waals and electrostatic interactions).

## 3.2. The role of colloidal stability in nanoparticle sedimentation–diffusion equilibrium

The effect of temperature on the sedimentation–diffusion equilibrium was evaluated by testing 100 nm diameter negatively charged gold nanoparticles in water at 23°C and at the biologically relevant

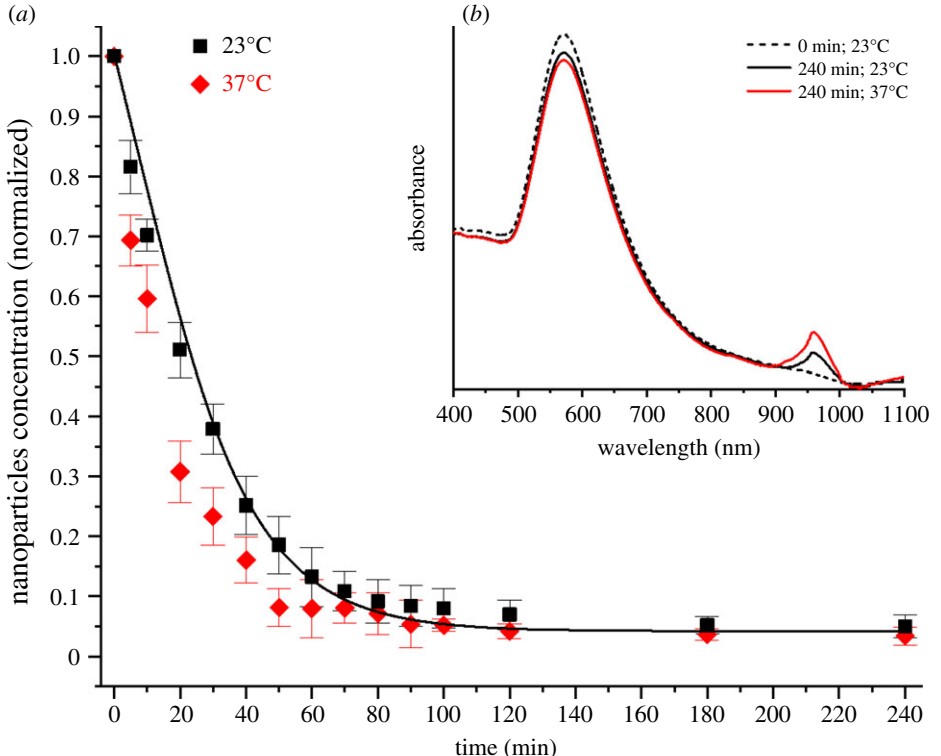

**Figure 7.** (*a*) Concentration of not-settled negatively charged 100 nm diameter gold nanoparticles in water at 23℃ (black squares) and 37℃ (red diamonds) over time together with predictions (lines) from the proposed modified Mason–Weaver model, and inset (*b*) UV–visible absorption spectra of nanoparticle solutions at the beginning of the experiment (dashed line) and after 240 min for solutions kept at 23℃ (black line) and at 37℃ (red line).

temperature of 37°C (figure 7*a*). The results obtained demonstrate that at 37°C the sedimentation velocity increased and that the solution reached the sedimentation–diffusion equilibrium 30 min before the same solution at 23°C. The higher temperature directly influences the aggregation kinetics of gold nanoparticles, enhancing their aggregation rate in solution [33]. Colloidal stability has been reported as one of the primary factors affecting the sedimentation–diffusion equilibrium of the nanoparticles [34]. The UV–visible spectra of the solutions kept at constant temperatures of 23°C and 37°C confirmed the increase in the aggregation rate of nanoparticles in solution with temperature (figure 7*b*). At higher temperatures aggregates of particles form at an increased rate and sediment faster than monodispersed nanoparticles in solution due their larger mass. The nanoparticle solution kept at 37°C exhibited a more evident depletion of the primary absorption peak and a higher secondary absorption peak when compared with the same solution kept at 23°C. The depletion of the primary absorption peak has been extensively reported as an indication of the reduced number of monodispersed nanoparticles in solution [35,36]. The depletion is caused by the particles and agglomerates precipitating to the bottom of the solution, outside of the scanning range of the spectrophotometer laser. The secondary absorption peak is likely to have been generated by clusters of nanoparticles formed in the solution and not yet part of the sediment. At the early stage of the aggregation process, nanoparticles have been reported to aggregate and assemble into specific structures, like dimers, trimers, tetramers, etc. [33]. Hence, these early-stage aggregation structures are likely to scatter light at longer wavelengths until they become part of the sediment at the bottom of the solution. Gold nanoparticles with ill-defined spherical geometry (like rods and stars) have been reported to exhibit their main absorption peak at wavelengths comparable with the wavelengths of our secondary absorption peak [37,38].

The comparison between the sedimentation–diffusion behaviour of oppositely charged 100 nm gold nanoparticles in water at 23°C is shown in figure 8*a*. There is no apparent difference between the sedimentation rate exhibited by these two types of nanoparticles; but positively charged nanoparticles exhibit a higher concentration of not-settled particles with respect to their negatively charged counterparts at equilibrium. This behaviour can be explained by considering the colloidal stability of the positively charged nanoparticle solution used in this investigation. The UV–visible analysis

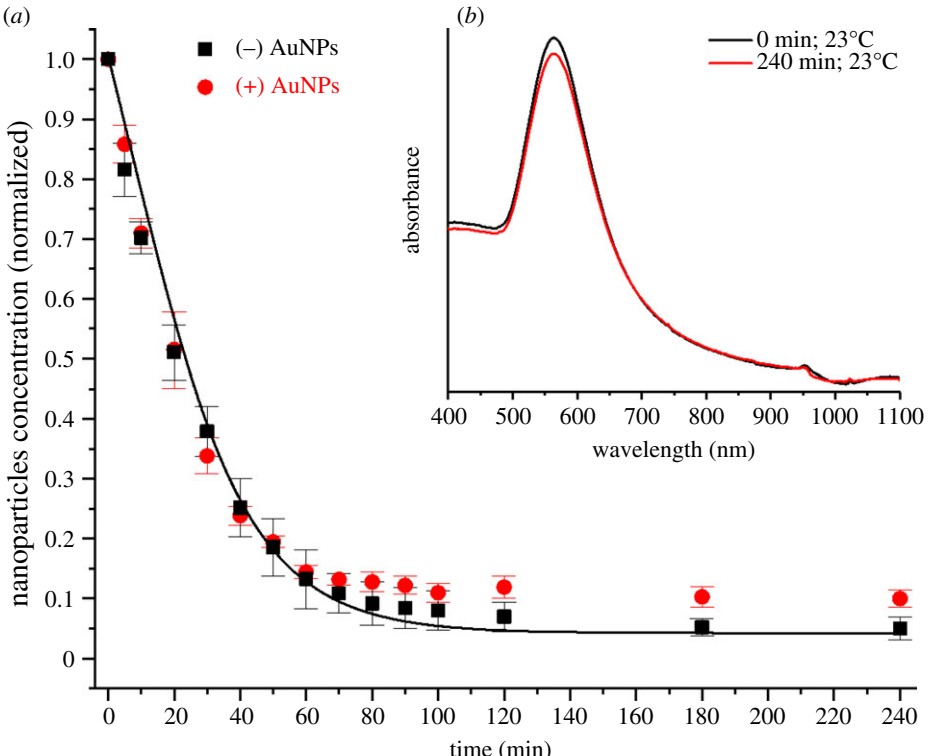

**Figure 8.** (*a*) Concentration of not-settled negatively charged (black squares) and positively charged (red circles) 100 nm diameter gold nanoparticles in water at 23°C together with predictions (lines) from the proposed modified Mason–Weaver model, and inset (*b*) UV–visible absorption spectra of positively charged nanoparticle solutions at the beginning of the experiment (black line) and after 240 min for solutions kept at 23°C (red line).

(figure 8*b*) confirmed the presence of fewer aggregates in solution at 23°C after 4 h when compared to the solution of negatively charged nanoparticles kept at the same temperature for the same time (i.e. less evident depletion of the primary absorption peak and no secondary absorption peak at longer wavelengths). Hence, more monodispersed particles are free to diffuse back into solution from the sediment leading to a higher concentration of not-settled nanoparticles at equilibrium.

## 3.3. Nanoparticle sedimentation–diffusion equilibrium in biological solutions

To characterize nanoparticle behaviour in biologically relevant solutions, the dynamics of sedimentation–diffusion equilibrium has been investigated by dispersing 100 nm diameter negatively charged gold nanoparticles in Dulbecco's modified Eagle medium (DMEM) with 10% fetal bovine serum (FBS). This solution was chosen because it is commonly used to support the growth of a range of mammalian cells and to understand the concentration of nanoparticles delivered at the cellular level. The potential presence of a two-dimensional adherent cell monolayer is expected to have an impact on the dynamics and concentration of nanoparticles in close proximity to the cell (i.e. nanoparticles already settled at the cellular level quantified in this investigation) due to uptake internalization as well as potential excretion by the cells.

Figure 9 shows the absence of significant deviations between the settling dynamics of nanoparticles in DMEM with 10% FBS and in water at 23°C and 37°C. This outcome can be explained considering the main factors affecting the settling behaviour of nanoparticles, namely the inter-particle interactions, the colloidal stability and the viscosity. A protein corona formed on the surface of the particles dispersed in the biological medium prevented nanoparticle aggregation that would otherwise be induced by the high content of salts in solution and the temperature [39]. The zeta potentials of the particles in the two solutions are similar [40,41], meaning that the electrostatic interactions have comparable magnitudes and result in no observable differences in the sedimentation profile. Finally, the dynamic viscosity of DMEM with 10% FBS has been reported to be only 5% higher than the dynamic viscosity of pure water [42], which is not enough to have a significant impact on nanoparticle kinetics [24].

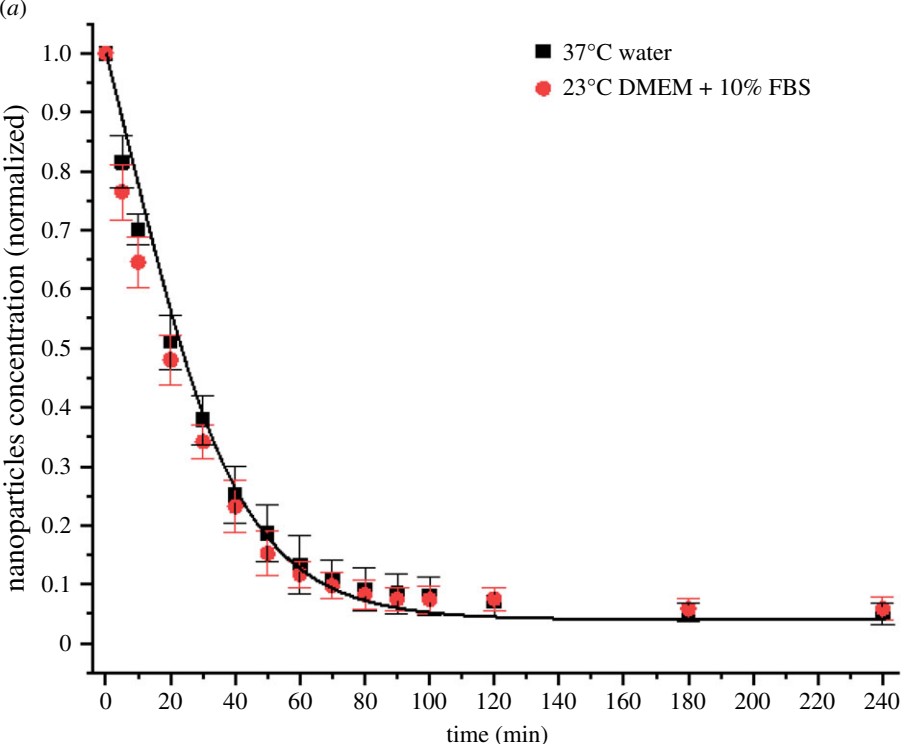

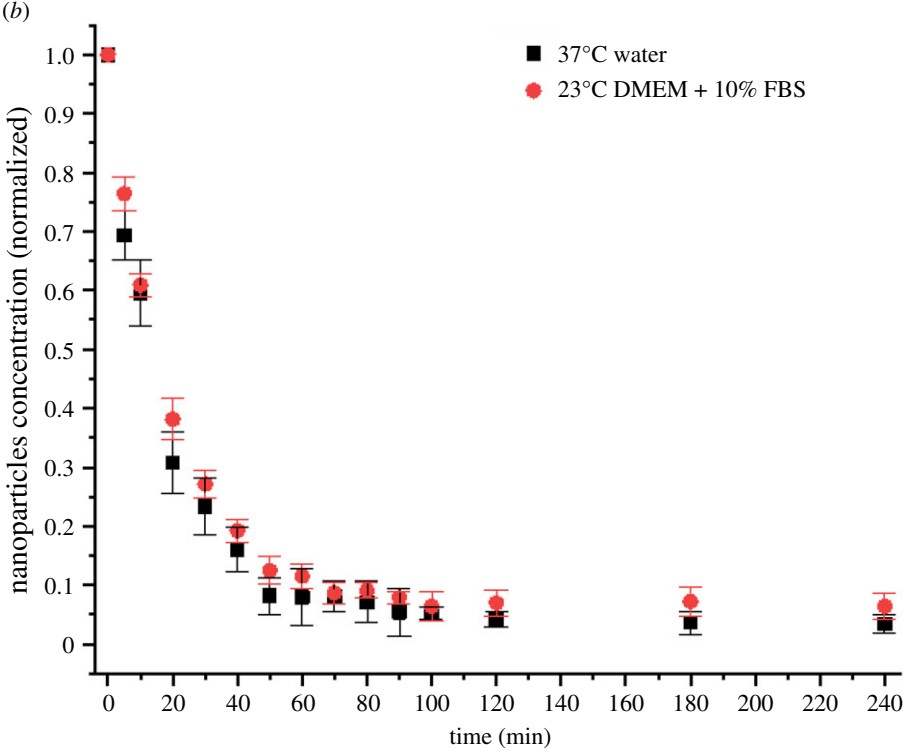

**Figure 9.** (*a*) Concentration of not-settled negatively charged 100 nm diameter gold nanoparticles in water (black squares) and in DMEM+10%FBS (red circles) as functions of time at 23℃ together with predictions (lines) from the proposed modified Mason–Weaver model and (*b*) concentration of the same nanoparticles dispersed in the same solutions at 37℃.

## 4. Conclusion

In this paper, we have presented an experimental investigation of the process of achieving sedimentation–diffusion equilibrium for nanoparticles in solution and developed a theoretical model

able to predict the settling dynamics of nanoparticles over time. The experimental results demonstrate that nanoparticle size, colloidal stability and solution temperature drive the sedimentation–diffusion equilibrium of nanoparticles in solution and that, for particles larger than 60 nm in diameter, the target dose differs significantly from the administered dose in both water and cell culture medium. This evidence should be taken into account when performing *in vitro* studies using adherent two-dimensional cultures, where the actual concentration of nanoparticles interacting with cells is needed to characterize and evaluate the cellular response. The theoretical model presented in this work shows good agreement with experimental data and is able to accurately describe the settling dynamics and concentration profile of nanoparticles in solution, making it a promising tool for the design of *in vitro* experiments and the study of concentration–response relationships.

Supporting information. Electronic supplementary material is available online.

Data accessibility. https://doi.org/10.5061/dryad.fn2z34trw.

Authors' contributions. F.G. conducted the experiments under the direct supervision of J.M.C. and E.A.P. F.G. and P.M. developed the mathematical model. M.W. and A.W. contributed to the design of the study and interpretation of the results. F.G. prepared the first draft of the manuscript and all authors contributed to the production of the final version. All authors have given approval to the final version of the manuscript.

Competing interests. The authors declare that they have no competing interests.

Funding. F.G. gratefully acknowledges funding from the UK Engineering and Physical Sciences Research Council in the form of a PhD studentship and funding from the European Commission Joint Research Centre (JRC) in the form of a traineeship.

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
