## [Peer Review File · Royal Society Open Science]

Review History

RSOS-201593.R0 (Original submission)

Review form: Reviewer 1

Is the manuscript scientifically sound in its present form?

No

Are the interpretations and conclusions justified by the results?

Yes

Is the language acceptable?

Yes

Do you have any ethical concerns with this paper?

No

Have you any concerns about statistical analyses in this paper?

No

Recommendation?

Reject

Comments to the Author(s)

First a note to the Editor. In the 21st century is it irritating to have to review a manuscript in old-fashioned format with, say, figures at the end. It's hard enough reviewing without having to move to and fro in the document. Although the paper itself gradually irritated me more and more, it was unfair to the authors to add to my irritation via an entirely unnecessary format. 95% of papers I review are now in an adequate layout making it easy to read figures etc. in context.

Now to the authors. In the academic game, making something about nano-tox (or nano-miracle cure) has been a reliable trick to increase the impact. This is despite the devastating review from Krug "Nanosafety Research – Are We on the Right Track?" where the answer, at least on tox, was an overwhelming "No". Of all the problems facing nano-bio, the idea that the dose might be erroneous because authors were too dumb to know that 100nm Au balls tend to fall out of solution either confirms Krug's view that many authors are, indeed, unsuited to do such work, or is an insult to the minority who aren't. In any case, the current authors are stacking the deck using the highest density nanoparticles most nano-bio users are likely to encounter.

If we strip out the pseudo-justification for the work, it comes down to saying that someone who thought that 100nm nano-Au would settle out in 2hrs because they used Mason-Weaver would be laughably wrong because in fact it takes 1hr.

As far as I know, when the Wert group (cited by this paper) did their Au settling experiments, their match against Mason-Weaver was rather good. I note that they carefully recorded (and controlled for) viscosity and had carefully measured the Au density. Neither value is mentioned in the paper or SI and after downloading 4.1GB of extra data, my Windows laptop was unable to open it so I have no idea if these elementary factors have been taken into consideration. (The word "density" appears only once in the m/s in a different context).

If you want to write a blockbuster expose of the failures of a 100-year old good-enough approximation then a few graphs of a hazily defined "not-settled" amount sitting above a "settled" amount isn't very convincing. Then there's Fig S1 which shows that all gold particles have the same low diffusion coefficient below some magic number then all instantly rise to Stokes-Einstein at 1.5 that number. This seems astonishing but the authors cite their own paper and indeed, my gut feel is wrong and there does seem to be an effect. In the one paper I went to, it was a doubling of the diffusion coefficient. In this paper it's an incredible (to me) 16x increase for the 60nm particles. There may be a reason for the discrepancy, but the authors don't bother to tell the poor reader about this.

[I was at first too lazy to read their other self-cited paper on these amazing effects. But to be scrupulously fair I finally opened it - to discover it says that the effect kicks in, independent of density in the 150-300nm regime, beyond the 60nm range where the current paper shows dramatic effects. And in any case whatever effects seemed to be a factor of 2, not 16. I really do not understand Fig S1 and the authors don't seem to have taken any trouble to explain it]

In any case, in an attempt to check out Mason-Weaver I visited a site that implements the theory (I searched Google for gravitational sedimentation calculator) and, as it happens, shows images of the Werts Au experiments. It takes not very long (though the plots are a bit confusing) to realise that a small error in density of the particles, or a poor estimation of viscosity is likely to be just as damaging to a nano-bio person as any error in diffusion coefficients of gold particles. In any case, down in more normal densities, presumably settling times are much longer and much more likely to be upset by, say, convective motion in the sample tubes.

In any case, the whole point of nano-bio is that the nano might interact with the bio in interesting ways. If you're going to put nano-Au into cultures with real cells (not just pseudo-bio DMEMS + !0% FBS), it might be that all sorts of bio and non-bio interactions will get in the way of the idealised settling. It was funny to read the sentence about the absence of a difference between water and the DMEMS/FBS. "This result, although interesting, is not surprising." If it's not

surprising, why is it interesting. Again, it's the danger of those trying to get all the credit of nano-bio without having to bother about the bio bit.

If I REALLY cared about nano-bio, I'd just say "Nanoparticles can settle so give your stuff a stir from time to time - oh, and look out for specific interactions with the biology, they might be interesting". That is far, far more valuable advice than this paper.

So. Given that the authors chose to write a nano-bio paper (Their opening sentence is "The biological response ..." then I reject the paper on the grounds that it's useless to the area where they claim that it's useful. If they want to submit a paper entitled "Mason and Weaver were wrong, and we can prove it" then either (a) those who seemed to have been OK with M/W for the past 100 years will be scratching their heads wondering how they missed such a massive error or (b) it's some minor effect that shows up only with gold (their 2017 paper with PS didn't show to me anything exciting) at exquisitely low concentrations, so who cares.

How six authors can measure a few settling times (admittedly with a technique I admire greatly) and collectively put together such an unconvincing paper will remain a mystery to me. Yes, they had to do some slightly more complex numerics, but I don't know what six people could contribute [This journal doesn't follow the practice of saying who did what for the paper].

Because the caustic method on these particles seems to me to be rather good (it may be boringly familiar or excitingly new, I've not checked), maybe the authors can do some real nano-bio and see in what ways their particles interact with real biology. I would be surprised if they had to apply their M/W correction as they'd almost certainly follow the good advice to shake/stir to avoid concentration gradients.

Review form: Reviewer 2

Is the manuscript scientifically sound in its present form?

Yes

Are the interpretations and conclusions justified by the results?

Yes

Is the language acceptable?

Yes

Do you have any ethical concerns with this paper?

No

Have you any concerns about statistical analyses in this paper?

No

Recommendation?

Major revision is needed (please make suggestions in comments)

Comments to the Author(s)

Giorgi et al. provide a combined modelling and experimental approach to examine the settling dynamics of nanoparticles. The modelling uses a modified form of the Mason-Weaver equation, a nonlinear advection-diffusion equation. The experimental approach involves the use of caustic signatures. The work provides interesting conclusions due to the reduced diffusivity of the nanoparticles. However, there are some improvements that could be made to this investigation.

The authors claim that they record caustic signatures at a separated distance of five microns. These measurements could then be used to construct a density profile of the nanoparticles as a

function of distance from the bottom of the well. This density profile is also the result of solving the Mason-Weaver equation. However, these observations are not included in the manuscript. The work would be much more convincing if these results were presented - instead of comparing the concentration of nanoparticles that are not settled, the observed density profile and the predicted density profile could be compared.

The authors select a Hill function as their choice of concentration-dependent nanoparticle diffusivity. This justification of this choice has not been discussed in the manuscript and should be included. Further, how were the values of the parameters in the Hill function selected (outside of D_{exp} and D_{StEin})?

The finite difference approximation used to obtain Equation (8) appears to use upwinding for the advection term (giving $n_{j+1} - n_j$) but makes the opposite choice for approximating the derivative in the diffusion function (giving $D(n_j) - D(n_{j-1})$). This seems an odd choice to me, given the flow is occurring in a single direction. The authors should justify this choice, or alternatively use consistent choices in their finite difference approximations.

The authors state that gravitational sedimentation starts to occur for particles with a diameter greater than 10 nm - this is incorrect, as gravity is a universal effect - it is more correct to state that sedimentary effects become relevant compared to diffusion effects.

In the authors model, they make the assumption that there is no flux through the bottom of the well. This is appropriate in the absence of cells, but various publications have demonstrated that the influence of the cells on the nanoparticle density profile is significant (see DeLoid et al, "Advanced computational modelling for in vitro nanomaterial dosimetry", Particle and Fibre Toxicology, 2015 and Faria et al, "Revisiting cell-particle association in vitro: a quantitative method to compare particle performance", Journal of Controlled Release, 2019). Given the authors have a modelling framework that can predict how much influence the reduced diffusivity has on the particle dosage, it would be interesting to see how much influence it has in the case where there are "cells" in the bottom of the dish.

Decision letter (RSOS-201593.R0)

Dear Dr giorgi

The Editors assigned to your paper RSOS-201593 "Settling dynamics of nanoparticles in simple and biological media" have made a decision based on their reading of the paper and any comments received from reviewers.

Regrettably, in view of the reports received, the manuscript has been rejected in its current form. However, a new manuscript may be submitted which takes into consideration these comments.

We invite you to respond to the comments supplied below and prepare a resubmission of your manuscript. Below the referees' and Editors' comments (where applicable) we provide additional requirements. We provide guidance below to help you prepare your revision.

Please note that resubmitting your manuscript does not guarantee eventual acceptance, and we do not generally allow multiple rounds of revision and resubmission, so we urge you to make every effort to fully address all of the comments at this stage. If deemed necessary by the Editors, your manuscript will be sent back to one or more of the original reviewers for assessment. If the original reviewers are not available, we may invite new reviewers.

Please resubmit your revised manuscript and required files (see below) no later than 24-May-2021. Note: the ScholarOne system will 'lock' if resubmission is attempted on or after this deadline. If you do not think you will be able to meet this deadline, please contact the editorial office immediately.

Please note article processing charges apply to papers accepted for publication in Royal Society Open Science (<https://royalsocietypublishing.org/rsos/charges>). Charges will also apply to papers transferred to the journal from other Royal Society Publishing journals, as well as papers submitted as part of our collaboration with the Royal Society of Chemistry (<https://royalsocietypublishing.org/rsos/chemistry>). Fee waivers are available but must be requested when you submit your manuscript (<https://royalsocietypublishing.org/rsos/waivers>).

Thank you for submitting your manuscript to Royal Society Open Science and we look forward to receiving your resubmission. If you have any questions at all, please do not hesitate to get in touch.

on behalf of Dr Robert Young (Associate Editor) and Miles Padgett (Subject Editor)
openscience@royalsociety.org

Associate Editor Comments to Author (Dr Robert Young):

Associate Editor: 1

Comments to the Author:

The reviewers have raised some serious concerns with the manuscript, which would require a complete overhaul to address. My recommendation is that you consider these carefully and resubmit a new manuscript if appropriate.

Reviewer comments to Author:

Reviewer: 1

Comments to the Author(s)

First a note to the Editor. In the 21st century is it irritating to have to review a manuscript in old-fashioned format with, say, figures at the end. It's hard enough reviewing without having to move to and fro in the document. Although the paper itself gradually irritated me more and more, it was unfair to the authors to add to my irritation via an entirely unnecessary format. 95% of papers I review are now in an adequate layout making it easy to read figures etc. in context. Now to the authors. In the academic game, making something about nano-tox (or nano-miracle cure) has been a reliable trick to increase the impact. This is despite the devastating review from Krug "Nanosafety Research – Are We on the Right Track?" where the answer, at least on tox, was an overwhelming "No". Of all the problems facing nano-bio, the idea that the dose might be

erroneous because authors were too dumb to know that 100nm Au balls tend to fall out of solution either confirms Krug's view that many authors are, indeed, unsuited to do such work, or is an insult to the minority who aren't. In any case, the current authors are stacking the deck using the highest density nanoparticles most nano-bio users are likely to encounter.

If we strip out the pseudo-justification for the work, it comes down to saying that someone who thought that 100nm nano-Au would settle out in 2hrs because they used Mason-Weaver would be laughably wrong because in fact it takes 1hr.

As far as I know, when the Wert group (cited by this paper) did their Au settling experiments, their match against Mason-Weaver was rather good. I note that they carefully recorded (and controlled for) viscosity and had carefully measured the Au density. Neither value is mentioned in the paper or SI and after downloading 4.1GB of extra data, my Windows laptop was unable to open it so I have no idea if these elementary factors have been taken into consideration. (The word "density" appears only once in the m/s in a different context).

If you want to write a blockbuster expose of the failures of a 100-year old good-enough approximation then a few graphs of a hazily defined "not-settled" amount sitting above a "settled" amount isn't very convincing. Then there's Fig S1 which shows that all gold particles have the same low diffusion coefficient below some magic number then all instantly rise to Stokes-Einstein at 1.5 that number. This seems astonishing but the authors cite their own paper and indeed, my gut feel is wrong and there does seem to be an effect. In the one paper I went to, it was a doubling of the diffusion coefficient. In this paper it's an incredible (to me) 16x increase for the 60nm particles. There may be a reason for the discrepancy, but the authors don't bother to tell the poor reader about this.

[I was at first too lazy to read their other self-cited paper on these amazing effects. But to be scrupulously fair I finally opened it - to discover it says that the effect kicks in, independent of density in the 150-300nm regime, beyond the 60nm range where the current paper shows dramatic effects. And in any case whatever effects seemed to be a factor of 2, not 16. I really do not understand Fig S1 and the authors don't seem to have taken any trouble to explain it]

In any case, in an attempt to check out Mason-Weaver I visited a site that implements the theory (I searched Google for gravitational sedimentation calculator) and, as it happens, shows images of the Werts Au experiments. It takes not very long (though the plots are a bit confusing) to realise that a small error in density of the particles, or a poor estimation of viscosity is likely to be just as damaging to a nano-bio person as any error in diffusion coefficients of gold particles. In any case, down in more normal densities, presumably settling times are much longer and much more likely to be upset by, say, convective motion in the sample tubes.

In any case, the whole point of nano-bio is that the nano might interact with the bio in interesting ways. If you're going to put nano-Au into cultures with real cells (not just pseudo-bio DMEMS + !0% FBS), it might be that all sorts of bio and non-bio interactions will get in the way of the idealised settling. It was funny to read the sentence about the absence of a difference between water and the DMEMS/FBS. "This result, although interesting, is not surprising." If it's not surprising, why is it interesting. Again, it's the danger of those trying to get all the credit of nano-bio without having to bother about the bio bit.

If I REALLY cared about nano-bio, I'd just say "Nanoparticles can settle so give your stuff a stir from time to time - oh, and look out for specific interactions with the biology, they might be interesting". That is far, far more valuable advice than this paper.

So. Given that the authors chose to write a nano-bio paper (Their opening sentence is "The biological response ..." then I reject the paper on the grounds that it's useless to the area where they claim that it's useful. If they want to submit a paper entitled "Mason and Weaver were wrong, and we can prove it" then either (a) those who seemed to have been OK with M/W for the past 100 years will be scratching their heads wondering how they missed such a massive error or (b) it's some minor effect that shows up only with gold (their 2017 paper with PS didn't show to me anything exciting) at exquisitely low concentrations, so who cares.

How six authors can measure a few settling times (admittedly with a technique I admire greatly) and collectively put together such an unconvincing paper will remain a mystery to me. Yes, they

had to do some slightly more complex numerics, but I don't know what six people could contribute [This journal doesn't follow the practice of saying who did what for the paper]. Because the caustic method on these particles seems to me to be rather good (it may be boringly familiar or excitingly new, I've not checked), maybe the authors can do some real nano-bio and see in what ways their particles interact with real biology. I would be surprised if they had to apply their M/W correction as they'd almost certainly follow the good advice to shake/stir to avoid concentration gradients.

Reviewer: 2

Comments to the Author(s)

Giorgi et al. provide a combined modelling and experimental approach to examine the settling dynamics of nanoparticles. The modelling uses a modified form of the Mason-Weaver equation, a nonlinear advection-diffusion equation. The experimental approach involves the use of caustic signatures. The work provides interesting conclusions due to the reduced diffusivity of the nanoparticles. However, there are some improvements that could be made to this investigation.

The authors claim that they record caustic signatures at a separated distance of five microns. These measurements could then be used to construct a density profile of the nanoparticles as a function of distance from the bottom of the well. This density profile is also the result of solving the Mason-Weaver equation. However, these observations are not included in the manuscript. The work would be much more convincing if these results were presented - instead of comparing the concentration of nanoparticles that are not settled, the observed density profile and the predicted density profile could be compared.

The authors select a Hill function as their choice of concentration-dependent nanoparticle diffusivity. This justification of this choice has not been discussed in the manuscript and should be included. Further, how were the values of the parameters in the Hill function selected (outside of D_{exp} and D_{StEin})?

The finite difference approximation used to obtain Equation (8) appears to use upwinding for the advection term (giving $n_{j+1} - n_j$) but makes the opposite choice for approximating the derivative in the diffusion function (giving $D(n_j) - D(n_{j-1})$). This seems an odd choice to me, given the flow is occurring in a single direction. The authors should justify this choice, or alternatively use consistent choices in their finite difference approximations.

The authors state that gravitational sedimentation starts to occur for particles with a diameter greater than 10 nm - this is incorrect, as gravity is a universal effect - it is more correct to state that sedimentary effects become relevant compared to diffusion effects.

In the authors model, they make the assumption that there is no flux through the bottom of the well. This is appropriate in the absence of cells, but various publications have demonstrated that the influence of the cells on the nanoparticle density profile is significant (see DeLoid et al, "Advanced computational modelling for in vitro nanomaterial dosimetry", Particle and Fibre Toxicology, 2015 and Faria et al, "Revisiting cell-particle association in vitro: a quantitative method to compare particle performance", Journal of Controlled Release, 2019). Given the authors have a modelling framework that can predict how much influence the reduced diffusivity has on the particle dosage, it would be interesting to see how much influence it has in the case where there are "cells" in the bottom of the dish.

===PREPARING YOUR MANUSCRIPT===

===PREPARING YOUR REVISION IN SCHOLARONE===

- An individual file of each figure (EPS or print-quality PDF preferred [either format should be produced directly from original creation package], or original software format).
- An editable file of each table (.doc, .docx, .xls, .xlsx, or .csv).
- An editable file of all figure and table captions.

- Any electronic supplementary material (ESM).
- If you are requesting a discretionary waiver for the article processing charge, the waiver form must be included at this step.
- If you are providing image files for potential cover images, please upload these at this step, and inform the editorial office you have done so. You must hold the copyright to any image provided.
- A copy of your point-by-point response to referees and Editors. This will expedite the preparation of your proof.

- Ensure that your data access statement meets the requirements at <https://royalsociety.org/journals/authors/author-guidelines/#data>. You should ensure that you cite the dataset in your reference list. If you have deposited data etc in the Dryad repository, please include both the 'For publication' link and 'For review' link at this stage.
- If you are requesting an article processing charge waiver, you must select the relevant waiver option (if requesting a discretionary waiver, the form should have been uploaded at Step 3 'File upload' above).
- If you have uploaded ESM files, please ensure you follow the guidance at <https://royalsociety.org/journals/authors/author-guidelines/#supplementary-material> to include a suitable title and informative caption. An example of appropriate titling and captioning may be found at https://figshare.com/articles/Table_S2_from_Is_there_a_trade-off_between_peak_performance_and_performance_breadth_across_temperatures_for_aerobic_scope_in_teleost_fishes_/3843624.

Author's Response to Decision Letter for (RSOS-201593.R0)

See Appendix A.

RSOS-210068.R0

Review form: Reviewer 1

Is the manuscript scientifically sound in its present form?

Yes

Are the interpretations and conclusions justified by the results?

Yes

Is the language acceptable?

Yes

Do you have any ethical concerns with this paper?

No

Have you any concerns about statistical analyses in this paper?

No

Recommendation?

Accept as is

Comments to the Author(s)

I thank the authors for their spirited defence of their work and their modifications to the paper.

I'm not at all persuaded by their response but at this stage, I really don't care. This is going to have no impact on anything. Those who really care about doing bio stuff with gold nanoparticles will be thinking through what happens as they're injected into the bloodstream or into a mass of cancer cells, and so won't (or shouldn't) be naively putting them into carefully quiescent tubes where the question of how much they do or do not settle onto the biological stuff at the bottom becomes relevant. Those who insist on doing careless and irrelevant experiments aren't going to read or respond to this paper. So it's all a gigantic waste of 6 researchers' efforts.

I'm still puzzled why no one else has spotted that Au nanoparticles have a 10x lower diffusion coefficient when dilute. Have no ultracentrifuge people never spotted this? But that's irrelevant to this paper.

So, given that the second reviewer was not as outraged as I was, if the Editor thinks it's OK, I really don't care.

Review form: Reviewer 2

Is the manuscript scientifically sound in its present form?

Yes

Are the interpretations and conclusions justified by the results?

Yes

Is the language acceptable?

Yes

Do you have any ethical concerns with this paper?

No

Have you any concerns about statistical analyses in this paper?

No

Recommendation?

Accept with minor revision (please list in comments)

Comments to the Author(s)

The authors have addressed the majority of my comments. I still have a couple of concerns:

The authors state that there is no need to consider how cells would impact the nanoparticle concentration as they do not measure in the final 125 microns. However, if the cells efficiently internalise/remove the particles from solution, the gradient will not build up in the same way. This should at least be discussed.

The authors also state that they do not report the concentration profiles as they cannot be measured in the bottom half of the solution. Even if the authors only reported the concentration profile in the top half of the solution this study would be much more convincing: it is much easier to obtain a model solution that matches a single observation (i.e. amount settled) than it is to match an entire concentration profile. It would be interesting to see whether the same reduction in diffusivity is present if the concentration profiles from Mason-Weaver are fit to the concentration profiles in the top half of the solution.

Review form: Reviewer 3

Is the manuscript scientifically sound in its present form?

Yes

Are the interpretations and conclusions justified by the results?

No

Is the language acceptable?

Yes

Do you have any ethical concerns with this paper?

No

Have you any concerns about statistical analyses in this paper?

No

Recommendation?

Major revision is needed (please make suggestions in comments)

Comments to the Author(s)

See attached file for comments (Appendix B).

Decision letter (RSOS-210068.R0)

Dear Dr giorgi

The Editors assigned to your paper RSOS-210068 "Settling dynamics of nanoparticles in simple and biological media" have now received comments from reviewers and would like you to revise the paper in accordance with the reviewer comments and any comments from the Editors. Please note this decision does not guarantee eventual acceptance.

Please submit your revised manuscript and required files (see below) no later than 21 days from today's (ie 18-May-2021) date. Note: the ScholarOne system will 'lock' if submission of the revision is attempted 21 or more days after the deadline. If you do not think you will be able to meet this deadline please contact the editorial office immediately.

on behalf of Dr Robert Young (Associate Editor) and Miles Padgett (Subject Editor)
openscience@royalsociety.org

Associate Editor Comments to Author (Dr Robert Young):

Associate Editor

Comments to the Author:

Apologies that it has taken some time to complete this review cycle; I felt it necessary to request a third reviewer. They have looked at the manuscript thoroughly and recommend a series of points be addressed, and changes are made before publication. I'd ask that you consider these carefully, please. I strongly believe that this process has strengthened the manuscript, and I hope you agree.

Reviewer comments to Author:

Reviewer: 1

Comments to the Author(s)

I thank the authors for their spirited defence of their work and their modifications to the paper. I'm not at all persuaded by their response but at this stage, I really don't care. This is going to have no impact on anything. Those who really care about doing bio stuff with gold nanoparticles will be thinking through what happens as they're injected into the bloodstream or into a mass of cancer cells, and so won't (or shouldn't) be naively putting them into carefully quiescent tubes where the question of how much they do or do not settle onto the biological stuff at the bottom

becomes relevant. Those who insist on doing careless and irrelevant experiments aren't going to read or respond to this paper. So it's all a gigantic waste of 6 researchers' efforts.

I'm still puzzled why no one else has spotted that Au nanoparticles have a 10x lower diffusion coefficient when dilute. Have no ultracentrifuge people never spotted this? But that's irrelevant to this paper.

So, given that the second reviewer was not as outraged as I was, if the Editor thinks it's OK, I really don't care.

Reviewer: 2

Comments to the Author(s)

The authors have addressed the majority of my comments. I still have a couple of concerns:

The authors state that there is no need to consider how cells would impact the nanoparticle concentration as they do not measure in the final 125 microns. However, if the cells efficiently internalise/remove the particles from solution, the gradient will not build up in the same way. This should at least be discussed.

The authors also state that they do not report the concentration profiles as they cannot be measured in the bottom half of the solution. Even if the authors only reported the concentration profile in the top half of the solution this study would be much more convincing: it is much easier to obtain a model solution that matches a single observation (i.e. amount settled) than it is to match an entire concentration profile. It would be interesting to see whether the same reduction in diffusivity is present if the concentration profiles from Mason-Weaver are fit to the concentration profiles in the top half of the solution.

Reviewer: 3

Comments to the Author(s)

See attached file for comments.

===PREPARING YOUR MANUSCRIPT===

===PREPARING YOUR REVISION IN SCHOLARONE===

Author's Response to Decision Letter for (RSOS-210068.R0)

See Appendix C.

RSOS-210068.R1

Review form: Reviewer 2

Is the manuscript scientifically sound in its present form?

Yes

Are the interpretations and conclusions justified by the results?

Yes

Is the language acceptable?

Yes

Do you have any ethical concerns with this paper?

No

Have you any concerns about statistical analyses in this paper?

No

Recommendation?

Accept as is

Comments to the Author(s)

The authors have addressed my comments.

Decision letter (RSOS-210068.R1)

Dear Dr Giorgi,

It is a pleasure to accept your manuscript entitled "Settling dynamics of nanoparticles in simple and biological media" in its current form for publication in Royal Society Open Science. The comments of the reviewer(s) who reviewed your manuscript are included at the foot of this letter.

on behalf of Dr Robert Young (Associate Editor) and Miles Padgett (Subject Editor)
openscience@royalsociety.org

Reviewer comments to Author:

Reviewer: 2

Comments to the Author(s)

The authors have addressed my comments.

Appendix A

We are grateful to the reviewers for their comments which helped us to improve the quality of the work presented. We have endeavoured to address all of the issues raised with a response below and changes to the manuscript which are highlighted in yellow.

Reviewer: 1

Comments to the Author(s)

First a note to the Editor. In the 21st century is it irritating to have to review a manuscript in old-fashioned format with, say, figures at the end. It's hard enough reviewing without having to move to and fro in the document. Although the paper itself gradually irritated me more and more, it was unfair to the authors to add to my irritation via an entirely unnecessary format. 95% of papers I review are now in an adequate layout making it easy to read figures etc. in context.

Now to the authors. In the academic game, making something about nano-tox (or nano-miracle cure) has been a reliable trick to increase the impact. This is despite the devastating review from Krug "Nanosafety Research—Are We on the Right Track?" where the answer, at least on tox, was an overwhelming "No". Of all the problems facing nano-bio, the idea that the dose might be erroneous because authors were too dumb to know that 100nm Au balls tend to fall out of solution either confirms Krug's view that many authors are, indeed, unsuited to do such work, or is an insult to the minority who aren't.

- *Krug suggested that the future nanotoxicological studies should include, as a requirement, a "consideration of the appropriate dose and/or concentration and the inclusion of a dose-effect relationship in the study design" [1]. Hence, we proposed a model to predict the concentration of nanomaterial delivered to the cellular level, thus providing a tool to better design toxicological investigations and to obtain more meaningful results. Moreover, our study does not have implications only for toxicology but in any investigation where an accurate dose of material interacting with cells is needed to correctly characterise any biological reaction. We have amended the introduction to include reference to Krug's review.*

In any case, the current authors are stacking the deck using the highest density nanoparticles most nano-bio users are likely to encounter.

- *We chose to use gold nanoparticles because they are well-known in the scientific community with a large number of papers proposing them as a promising carrier for the targeted delivery of therapeutic, diagnostic and imaging agents in the human body. The tracking technique employed has been previously demonstrated for particles of lower material density and hence the model described in the manuscript could be easily applied to these particles. We have included a statement to this effect in the introduction.*

If we strip out the pseudo-justification for the work, it comes down to saying that someone who thought that 100nm nano-Au would settle out in 2hrs because they used Mason-Weaver would be laughably wrong because in fact it takes 1hr.

- *We have addressed the reviewer's comments about the justification for our study above. Our conclusions relate to both the time to reach sedimentation-diffusion equilibrium AND the concentration of particles delivered at the cellular level. The settling time can be directly related to the sedimentation dynamics of the particles, which has been reported to be a primary factor regulating cellular uptakes and toxicity [2]. We have amended the discussion and conclusions to emphasis these points.*

As far as I know, when the Wert group (cited by this paper) did their Au settling experiments, their match against Mason-Weaver was rather good. I note that they carefully recorded (and controlled for) viscosity and had carefully measured the Au density. Neither value is mentioned in the paper or SI and after downloading 4.1GB of extra data, my Windows laptop was unable to open it so I have no idea if these elementary factors have been taken into consideration. (The word "density" appears only once in the m/s in a different context).

- *We have added a discussion about the viscosity which has been reported previously to be not influenced by the concentration of particles used in this study. It would appear, based on the cited work by the Wert group that the reviewer uses "density" where we have used "concentration of nanoparticles" expressed in ml^{-1} . We extensively investigated, and discussed in the manuscript, the concentration gradient of nanoparticles in solution in our manuscript. Our working concentration range is biologically relevant and used by a number of studies in the literature (for example [3-7]).*

If you want to write a blockbuster expose of the failures of a 100-year old good-enough approximation then a few graphs of a hazily defined "not-settled" amount sitting above a "settled" amount isn't very convincing.

- *We do not claim to have written a 'blockbuster exposé'. We are reporting our scientifically rigorous attempts to resolve a problem that is ill-defined because there is no definitive demarcation between settled and not-settled – the boundary is fuzzy by definition. The term "settled" is commonly used in the scientific community to identify particles forming a sediment. We have extended our discussion of this issue in the manuscript.*

Then there's Fig S1 which shows that all gold particles have the same low diffusion coefficient below some magic number then all instantly rise to Stokes-Einstein at 1.5 that number. This seems astonishing but the authors cite their own paper and indeed, my gut feel is wrong and there does seem to be an effect. In the one paper I went to, it was a doubling of the diffusion coefficient. In this paper it's an incredible (to me) 16x increase for the 60nm particles. There may be a reason for the discrepancy, but the authors don't bother to tell the poor reader about this.

[I was at first too lazy to read their other self-cited paper on these amazing effects. But to be scrupulously fair I finally opened it - to discover it says that the effect kicks in, independent of density in the 150-300nm regime, beyond the 60nm range where the current paper shows dramatic effects. And in any case whatever effects seemed to be a factor of 2, not 16. I really do not understand Fig S1 and the authors don't seem to have taken any trouble to explain it].

- *Since figure S1 seems to have confused rather than clarified the issues, we have removed it from the supplementary material and instead referred to the original results. However, we feel we should clarify that in the earlier work by Coglitore et al [8], the measured diffusion coefficients did not change significantly with concentration for gold particles of diameter 10 nm to 150 nm in water and, in this size range, were several orders of magnitude smaller than predicted by the Stokes-Einstein relationship.*

In any case, in an attempt to check out Mason-Weaver I visited a site that implements the theory (I searched Google for gravitational sedimentation calculator) and, as it happens, shows images of the Werts Au experiments. It takes not very long (though the plots are a bit confusing) to realise that a small error in density of the particles, or a poor estimation of viscosity is likely to be just as damaging to a nano-bio person as any error in diffusion coefficients of gold particles. In any case, down in more normal densities, presumably settling times are much longer and much more likely to be upset by, say, convective motion in the sample tubes.

- *We have included a discussion of the sensitivity of the Mason-Weaver model to viscosity of the medium and the concentration of the particles. Please see our comment above about the relevance of our concentrations and densities.*

In any case, the whole point of nano-bio is that the nano might interact with the bio in interesting ways. If you're going to put nano-Au into cultures with real cells (not just pseudo-bio DMEM + 10% FBS), it might be that all sorts of bio and non-bio interactions will get in the way of the idealised settling. It was funny to read the sentence about the absence of a difference between water and the DMEM/FBS. "This result, although interesting, is not surprising." If it's not surprising, why is it interesting. Again, it's the

danger of those trying to get all the credit of nano-bio without having to bother about the bio bit.

- *We have amended the sentence to which the reviewer has objected and provided a stronger rationale for our comments.*

If I REALLY cared about nano-bio, I'd just say "Nanoparticles can settle so give your stuff a stir from time to time - oh, and look out for specific interactions with the biology, they might be interesting". That is far, far more valuable advice than this paper.

- *The primary focus of any investigation of the bio-nano interface is to evaluate the response of a biological organism following the exposure to a nanomaterial. Hence, preventing the nanomaterial settling and/or altering the concentration of nanomaterial at the cellular level by shaking or stirring is usually counterproductive. The purpose of this study was to characterise the transport of nanomaterial throughout the solution and provide an estimation of the amount delivered to cellular level, not to monitor or quantify the bio-nano interaction.*

So. Given that the authors chose to write a nano-bio paper (Their opening sentence is "The biological response ..." then I reject the paper on the grounds that it's useless to the area where they claim that it's useful. If they want to submit a paper entitled "Mason and Weaver were wrong, and we can prove it" then either (a) those who seemed to have been OK with M/W for the past 100 years will be scratching their heads wondering how they missed such a massive error or (b) it's some minor effect that shows up only with gold (their 2017 paper with PS didn't show to me anything exciting) at exquisitely low concentrations, so who cares.

- *We would like to refute these criticisms. Our motivation in conducting the study was to make direct real-time measurements of the sedimentation of nanoparticles for the first time. The discovery that the Mason-Weaver was inaccurate was an outcome and was unlikely to have been observed in the past because the technology was unavailable for direct real-time measurements. The differences between the Mason-Weaver predictions and our measurements are of the order of 30% and hence cannot be considered minor, especially given that metallic nanoparticles are encountered in a number of biological scenarios where it is important to understand their interaction with cells. As mentioned above, our concentration ranges are biologically relevant and used by a number of studies in the literature (for example [2-6]). We have expanded the discussion of these issues in the manuscript.*

How six authors can measure a few settling times (admittedly with a technique I admire

greatly) and collectively put together such an unconvincing paper will remain a mystery to me. Yes, they had to do some slightly more complex numerics, but I don't know what six people could contribute [This journal doesn't follow the practice of saying who did what for the paper].

- *We have added an authors' contribution section in the manuscript.*

Because the caustic method on these particles seems to me to be rather good (it may be boringly familiar or excitingly new, I've not checked), maybe the authors can do some real nano-bio and see in what ways their particles interact with real biology. I would be surprised if they had to apply their M/W correction as they'd almost certainly follow the good advice to shake/stir to avoid concentration gradients.

- *We are undertaking experiments with caustics to investigate the interaction of nanoparticles with cells and hope to report it in the future; however, we are not shaking or stirring the solutions because that would prevent the very interactions whose mechanisms we wish to study. Hence, we need to consider the concentration gradients that are the focus of this manuscript.*

Reviewer: 2

Comments to the Author(s)

Giorgi et al. provide a combined modelling and experimental approach to examine the settling dynamics of nanoparticles. The modelling uses a modified form of the Mason-Weaver equation, a nonlinear advection-diffusion equation. The experimental approach involves the use of caustic signatures. The work provides interesting conclusions due to the reduced diffusivity of the nanoparticles. However, there are some improvements that could be made to this investigation.

The authors claim that they record caustic signatures at a separated distance of five microns. These measurements could then be used to construct a density profile of the nanoparticles as a function of distance from the bottom of the well. This density profile is also the result of solving the Mason-Weaver equation. However, these observations are not included in the manuscript. The work would be much more convincing if these results were presented - instead of comparing the concentration of nanoparticles that are not settled, the observed density profile and the predicted density profile could be compared.

- *Yes, we agree with the reviewer; however, it is a limitation of our technique that at high concentrations of particles the caustics overlap making our measurements less precise and unreliable. Hence, we did not provide the density profiles as the main results of our investigation because we are not able to accurately count the*

number of particles in the bottom half of the solution once the equilibrium is attained. We have added this explanation to the manuscript.

The authors select a Hill function as their choice of concentration-dependent nanoparticle diffusivity. This justification of this choice has not been discussed in the manuscript and should be included. Further, how were the values of the parameters in the Hill function selected (outside of D_{exp} and D_{StEin})?

- *We provided a further discussion and clarification in the manuscript.*

The finite difference approximation used to obtain Equation (8) appears to use upwinding for the advection term (giving $n_{\{j+1\}} - n_{\{j\}}$) but makes the opposite choice for approximating the derivative in the diffusion function (giving $D(n_{\{j\}}) - D(n_{\{j-1\}})$). This seems an odd choice to me, given the flow is occurring in a single direction. The authors should justify this choice, or alternatively use consistent choices in their finite difference approximations.

- *We agree and have updated the finite difference approximation for the diffusion function as $D(n_{\{j+1\}}) - D(n_{\{j\}})$. We have updated figure 3 where it makes negligible difference to the predictions and has no impact on the conclusions.*

The authors state that gravitational sedimentation starts to occur for particles with a diameter greater than 10 nm - this is incorrect, as gravity is a universal effect - it is more correct to state that sedimentary effects become relevant compared to diffusion effects.

- *To remove the ambiguity of this phrase the sentence has been modified as follows:
"Gravitational sedimentation becomes relevant for particles with a diameter larger than 10 nm, causing a concentration gradient from the bottom to the top of the solution"*

In the authors model, they make the assumption that there is no flux through the bottom of the well. This is appropriate in the absence of cells, but various publications have demonstrated that the influence of the cells on the nanoparticle density profile is significant (see DeLoid et al, "Advanced computational modelling for in vitro nanomaterial dosimetry", Particle and Fibre Toxicology, 2015 and Faria et al, "Revisiting cell-particle association in vitro: a quantitative method to compare particle performance", Journal of Controlled Release, 2019). Given the authors have a modelling framework that can predict how much influence the reduced diffusivity has on the particle dosage, it would be interesting to see how much influence it has in the case where there are "cells" in the bottom of the dish.

- *Our observation starts at 125 μ m which is relatively far from the bottom of the sample, for that reason we believe that the presence of cells is unlikely to cause an evident alteration of the dynamics of the nanoparticles in that part of the*

solution. The purpose of this study is to characterise the transport of nanomaterial throughout the solution and provide an estimation of the amount delivered to cellular level, not to monitor/quantify the bio-nano interaction.

References

- [1] Krug HF. Nanosafety research-are we on the right track? *Angew Chem Int Ed* 2014;53(46):12304-12319.
- [2] Cho EC, Zhang Q, Xia Y. The effect of sedimentation and diffusion on cellular uptake of gold nanoparticles. *Nat Nanotechnol* 2011;6(6):385-391.
- [3] Rizk N, Christoforou N, Lee S. Optimization of anti-cancer drugs and a targeting molecule on multifunctional gold nanoparticles. *Nanotechnology* 2016;27(18).
- [4] Vedantam P, Tzeng T-J, Brown AK, Podila R, Rao A, Staley K. Binding of Escherichia coli to Functionalized Gold Nanoparticles. *Plasmonics* 2012;7(2):301-308.
- [5] Vedantam P, Huang G, Tzeng TRJ. Size-dependent cellular toxicity and uptake of commercial colloidal gold nanoparticles in DU-145 cells. *Cancer Nanotechnol* 2013;4(1-3):13-20.
- [6] Puvanakrishnan P, Park J, Chatterjee D, Krishnan S, Tunnell JW. In vivo tumor targeting of gold nanoparticles: Effect of particle type and dosing strategy. *Int J Nanomed* 2012;7:1251-1258.
- [7] Hinderliter PM, Minard KR, Orr G, Chrisler WB, Thrall BD, Pounds JG, et al. ISDD: A computational model of particle sedimentation, diffusion and target cell dosimetry for in vitro toxicity studies. *Part Fibre Toxicol* 2010;7.
- [8] Coglitore D, Edwardson SP, Macko P, Patterson EA, Whelan M. Transition from fractional to classical Stokes-Einstein behaviour in simple fluids. *R Soc Open Sci* 2017;4(12).

Appendix B

Experimental data of the settling dynamics of nanoparticles is first presented using caustic signatures and then a theoretical model is developed accordingly. In this way, interesting conclusions related to differences in the diffusivity of the nanoparticles are provided. However, the model is not used or discussed in the subsequent sections where more realistic conditions are tested and differences in the sedimentation profile are observed. Despite the insistence of the authors about the importance of the paper for the nano-bio community, the nano-bio work is limited to one inconclusive paragraph in section 2.3. The following points should be addressed:

In page 3 line 17, the authors state that the aim of the study is to provide a better understanding of the dose of nanoparticles delivered to the cellular level. There is a huge gap between nanoparticle cell delivery and the experiments done by the authors, therefore I suggest rephrasing this sentence. “the sedimentation profile of nanoparticles in simple and biological media” or “the local concentration of nanoparticles to which the cells are actually exposed” better describes what is done in this work.

Page 3 line 40. Gravitational sedimentation become relevant for particles with a diameter bigger than 10 nm. This may be true for the case of gold but e.g. silica or polystyrene nanoparticles do not sediment to much larger sizes. Please correct the sentence accordingly.

Page 6 line 24. The critical size and the concentration values should be explicitly mentioned in the sentence for the sake of the reader. In the same sentence, a good practice when referring to self-citations is the use of expressions like “In our previous work”, particularly if these are used as an argument to support author ideas. On one hand, this indicates previous experience in the field. In the other hand, it helps to the reader to evaluate the strength of the argument used.

Section 2.2 can be summarized as: increasing the temperature, promotes aggregation of nanoparticles that in turns speed up sedimentation. Ah! And positive nanoparticles are less stable. I am sure this is case-specific and not always true. However, nothing is mentioned about the surfactant on the surface of the particles and their surface charge, and the pH of the media. These parameters should be specified and discussed since they strongly influence and determine the observed results.

Then, the authors buy a bottle of gold nanoparticles that have been stable for weeks. They dilute some millilitres in water and suddenly... the particles start aggregating?! Which is the reason, depletion of surfactant, change in the pH?

Page 9 line 3. The UV-vis aggregation profile from Figure 5b seems strange to me and differs from those aggregation profiles shown in refs 33, 35 and 36. Usually, when citrate particles aggregate (I assume citrate from the company’s webpage) the dipole peak decreases in intensity as it shifts to the red and widens. The intensity at longer wavelengths increases but not as a well-defined narrow peak. The authors should provide further evidences that support their thoughts about aggregation.

Figure 5. After 240 min the concentration of nanoparticles in solution is 10 times lower than the initial concentration (Figure 5a) but the UV-vis spectra (Figure 5b) show almost the same intensity. Have the UV-vis spectra been normalized? Which fraction of the solution have been taken, the non-sedimented? A fraction that contains also sediments? Please clarify.

Page 10 line 17. The authors state that the complex gold-protein NPs exhibit the same zeta potential of the same bare NPs. This may be true in a very few cases, but it is the exception rather than the rule (see e.g. Eudald et al. ACS Nano 2010, 4, 7, 3623–3632). Therefore, the authors should provide their own zeta potential measurements, instead of just referring to works where different gold nanoparticles and concentrations may be used. Why -20 mV and -40 mV are comparable magnitudes when a commonly accepted limit for electrostatic stabilization is -30 mV? Below that value particles

can not be stabilized electrostatically and thus they fast aggregate if they are not sterically protected. Then, let's assume that the zeta potential of citrate particles in water is -35 mV and -25 mV with protein coating in water (but they are in DMEM!, which is a very high ionic strength media that screen the surface charge, and the zeta potential is probably much lower). Isn't this a very huge difference to simply say, ok because they have the same zeta potential, we observe no differences in the sedimentation profile?

The authors develop a model in section 2.1 that can accurately describe particle transport in solution, but it is not mentioned or used in sections 2.2 and 2.3. Then, what is the link between section 2.1 with the others? Does the model work for 37C and for positive particles?

Appendix C

Authors' Responses to Reviewers' comments:

We acknowledge the reviewers for their comments which helped us to increase the quality of the work presented. We have endeavoured to address all of the issues raised with a response below and changes to the manuscript which are highlighted in yellow.

Reviewer: #1

Comments to the Author(s)

I thank the authors for their spirited defence of their work and their modifications to the paper.

I'm not at all persuaded by their response but at this stage, I really don't care. This is going to have no impact on anything. Those who really care about doing bio stuff with gold nanoparticles will be thinking through what happens as they're injected into the bloodstream or into a mass of cancer cells, and so won't (or shouldn't) be naively putting them into carefully quiescent tubes where the question of how much they do or do not settle onto the biological stuff at the bottom becomes relevant. Those who insist on doing careless and irrelevant experiments aren't going to read or respond to this paper. So it's all a gigantic waste of 6 researchers' efforts.

I'm still puzzled why no one else has spotted that Au nanoparticles have a 10x lower diffusion coefficient when dilute. Have no ultracentrifuge people never spotted this? But that's irrelevant to this paper.

So, given that the second reviewer was not as outraged as I was, if the Editor thinks it's OK, I really don't care.

- *We acknowledge Reviewer 1 for the comments. However, given the lack of comments relevant to our paper or to our previous responses we have focussed our attention on the comments and feedback provided by Reviewer 2 and Reviewer 3. We would like only to point out again that the above so-called "irrelevant experiments" are the standard for in-vitro investigations involving nanoparticles and biological organisms.*

Reviewer: #2

Comments to the Author(s)

The authors have addressed the majority of my comments. I still have a couple of concerns: The authors state that there is no need to consider how cells would impact the nanoparticle concentration as they do not measure in the final 125 microns. However, if the cells efficiently internalise/remove the particles from solution, the gradient will not build up in the same way. This should at least be discussed.

- *We have added a short discussion about the potential influence of a 2D adherent cell culture to the manuscript.*

The authors also state that they do not report the concentration profiles as they cannot be measured in the bottom half of the solution. Even if the authors only reported the concentration profile in the top half of the solution this study would be much more convincing: it is much easier to obtain a model solution that matches a single observation (i.e. amount settled) than it is to match an entire concentration profile. It would be interesting to see whether the same reduction in diffusivity is present if the concentration profiles from Mason-Weaver are fit to the concentration profiles in the top half of the solution.

- *We have added to the manuscript the experimental concentration profile fitted with the theoretical concentration profile as evaluated by the modified version of the Mason – Weaver model.*

Reviewer: 3

Comments to the Author(s)

Experimental data of the settling dynamics of nanoparticles is first presented using caustic signatures and then a theoretical model is developed accordingly. In this way, interesting conclusions related to differences in the diffusivity of the nanoparticles are provided. However, the model is not used or discussed in the subsequent sections where more realistic conditions are tested and differences in the sedimentation profile are observed. Despite the insistence of the authors about the importance of the paper for the nano-bio community, the nano-bio work is limited to one inconclusive paragraph in section 2.3. The following points should be addressed:

In page 3 line 17, the authors state that the aim of the study is to provide a better understanding of the dose of nanoparticles delivered to the cellular level. There is a huge gap between nanoparticle cell delivery and the experiments done by the authors, therefore I suggest rephrasing this sentence. “the sedimentation profile of nanoparticles in simple and biological media” or “the local concentration of nanoparticles to which the cells are actually exposed” better describes what is done in this work.

- *We agree and have changed the aim in the way suggested.*

Page 3 line 40. Gravitational sedimentation become relevant for particles with a diameter bigger than 10 nm. This may be true for the case of gold but e.g. silica or polystyrene nanoparticles do not sediment to much larger sizes. Please correct the sentence accordingly.

- *We agree about being more specific and have modified the manuscript accordingly.*

Page 6 line 24. The critical size and the concentration values should be explicitly mentioned in the sentence for the sake of the reader. In the same sentence, a good practice when referring to selfcitations is the use of expressions like “In our previous work”, particularly if these are used as an argument to support author ideas. On one hand, this indicates previous experience in the field. In the other hand, it helps to the reader to evaluate the strength of the argument used.

- *We agree with the reviewer and have provided information about the critical values of size and concentration in the manuscript.*

Section 2.2 can be summarized as: increasing the temperature, promotes aggregation of nanoparticles that in turns speed up sedimentation. Ah! And positive nanoparticles are less stable. I am sure this is case-specific and not always true. However, nothing is mentioned about the surfactant on the surface of the particles and their surface charge, and the pH of the media. These parameters should be specified and discussed since they strongly influence and determine the observed results. Then, the authors buy a bottle of gold nanoparticles that have been stable for weeks. They dilute some millilitres in water and suddenly... the particles start aggregating?! Which is the reason, depletion of surfactant, change in the pH?

- *We have specified the surfactant used to stabilise the nanoparticle and the pH of the stock solution in the material and method section of the manuscript. We have also moved the material and method section of the manuscript before the results and discussion section to avoid any confusion. We have added a significant further explanation in the manuscript about the reason for the aggregation of the nanoparticles.*

Page 9 line 3. The UV-vis aggregation profile from Figure 5b seems strange to me and differs from those aggregation profiles shown in refs 33, 35 and 36. Usually, when citrate particles aggregate (I assume citrate from the company's webpage) the dipole peak decreases in intensity as it shifts to the red and widens. The intensity at longer wavelengths increases but not as a well-defined narrow peak. The authors should provide further evidences that support their thoughts about aggregation.

- *We have added a significant further explanation in the manuscript.*

Figure 5. After 240 min the concentration of nanoparticles in solution is 10 times lower than the initial concentration (Figure 5a) but the UV-vis spectra (Figure 5b) show almost the same intensity. Have the UV-vis spectra been normalized? Which fraction of the solution have been taken, the nonsedimented? A fraction that contains also sediments? Please clarify.

- *The UV-Vis spectra have not been normalised. The spectroscopic analysis used 3 ml of the solution in a glass cuvette of depth 45 mm. On the other hand, the sedimentation was performed with 60 μ l of solution in a cavity of microscopy slide 250 μ m \pm 10 μ m deep. Hence, it is inappropriate to compare the concentration of nanoparticles sedimented in the two systems. We have added this experimental detail in the material and methods section of the manuscript.*

Page 10 line 17. The authors state that the complex gold-protein NPs exhibit the same zeta potential of the same bare NPs. This may be true in a very few cases, but it is the exception rather than the rule (see e.g. Eudald et al. ACS Nano 2010, 4, 7, 3623–3632). Therefore, the authors should provide their own zeta potential measurements, instead of just referring to works where different gold nanoparticles and concentrations may be used. Why -20 mV and -40 mV are comparable magnitudes when a commonly accepted limit for electrostatic stabilization is -30 mV? Below that value particles cannot be stabilized electrostatically and thus they fast aggregate if they are not sterically protected. Then, let's assume that the zeta potential of citrate particles in water is -35 mV and -25 mV with protein coating in water (but they are in DMEM!, which is a very high ionic strength media that screen the surface charge, and the zeta potential is probably much lower). Isn't this a very huge difference to simply say, ok because they have the same zeta potential, we observe no differences in the sedimentation profile?

- *We do not have any measurements of zeta potentials and hence we are happy to accept the reviewer's advice and have replaced our statement with a simpler one based on the reviewer's suggestion.*

The authors develop a model in section 2.1 that can accurately describe particle transport in solution, but it is not mentioned or used in sections 2.2 and 2.3. Then, what is the link between section 2.1 with the others? Does the model work for 37°C and for positive particles?

- *We have accepted reviewer's suggestion and added the modified Mason-Weaver prediction at 23°C in the graphs related to sections 2.2 and 2.3. Concerning the temperature, regrettably we do not have experimental data for the diffusion coefficient of gold nanoparticles at 37 °C and we unable to test the model as suggested.*

References

- [1] Coglitore D, Edwardson SP, Macko P, Patterson EA, Whelan M. Transition from fractional to classical Stokes-Einstein behaviour in simple fluids. *R Soc Open Sci* 2017;4(12).
- [2] Dutta A, Paul A, Chattopadhyay A. The effect of temperature on the aggregation kinetics of partially bare gold nanoparticles. *RSC Adv* 2016;6(85):82138-82149.
- [3] Mehrdel B, Abdul Aziz A, Yoon TL, Lee SC. Effect of chemical interface damping and aggregation size of bare gold nanoparticles in NaCl on the plasmon resonance damping. *Opt Mater Express* 2017;7(3):955-966.
- [4] Giorgi-Coll S, Marín MJ, Sule O, Hutchinson PJ, Carpenter KLH. Aptamer-modified gold nanoparticles for rapid aggregation-based detection of inflammation: an optical assay for interleukin-6. *Microchim Acta* 2020;187(1).
- [5] Fathi F, Rashidi M-, Omid Y. Ultra-sensitive detection by metal nanoparticles-mediated enhanced SPR biosensors. *Talanta* 2019;192:118-127.
- [6] Kumar R, Binetti L, Nguyen TH, Alwis LSM, Agrawal A, Sun T, et al. Determination of the Aspect-ratio Distribution of Gold Nanorods in a Colloidal Solution using UV-visible absorption spectroscopy. *Sci Rep* 2019;9(1).